# Metabolite interactions in the bacterial Calvin cycle and implications for flux regulation

Emil Sporre[1,5], Jan Karlsen[1,5], Karen Schriever [2], Johannes Asplund-Samuelsson[1], Markus Janasch[1,4], Linnéa Strandberg[1], Anna Karlsson[1], David Kotol [1], Luise Zeckey[1], Ilaria Piazza[3], Per-Olof Syrén[2], Fredrik Edfors[1] & Elton P. Hudson [1✉]

Metabolite-level regulation of enzyme activity is important for microbes to cope with environmental shifts. Knowledge of such regulations can also guide strain engineering for biotechnology. Here we apply limited proteolysis-small molecule mapping (LiP-SMap) to identify and compare metabolite-protein interactions in the proteomes of two cyanobacteria and two lithoautotrophic bacteria that fix $CO_2$ using the Calvin cycle. Clustering analysis of the hundreds of detected interactions shows that some metabolites interact in a species-specific manner. We estimate that approximately 35% of interacting metabolites affect enzyme activity in vitro, and the effect is often minor. Using LiP-SMap data as a guide, we find that the Calvin cycle intermediate glyceraldehyde-3-phosphate enhances activity of fructose-1,6/sedoheptulose-1,7-bisphosphatase (F/SBPase) from *Synechocystis* sp. PCC 6803 and *Cupriavidus necator* in reducing conditions, suggesting a convergent feed-forward activation of the cycle. In oxidizing conditions, glyceraldehyde-3-phosphate inhibits *Synechocystis* F/SBPase by promoting enzyme aggregation. In contrast, the glycolytic intermediate glucose-6-phosphate activates F/SBPase from *Cupriavidus necator* but not F/SBPase from *Synechocystis*. Thus, metabolite-level regulation of the Calvin cycle is more prevalent than previously appreciated.

[1] Department of Protein Science, Science for Life Laboratory, KTH - Royal Institute of Technology, Stockholm, Sweden. [2] Department of Fiber and Polymer Technology, Science for Life Laboratory, KTH - Royal Institute of Technology, Stockholm, Sweden. [3] Max Delbrück Center for Molecular Medicine in the Helmholtz Association, Berlin, Germany. [4]Present address: Department of Biotechnology and Nanomedicine, SINTEF Industry, 7465 Trondheim, Norway. [5]These authors contributed equally: Emil Sporre, Jan Karlsen. ✉email: paul.hudson@scilifelab.se

Metabolite-protein interactions, such as allosteric inhibition or activation of enzymes, can act as feedback mechanisms for adapting metabolic flux to changing conditions[1,2], and can also be targets for metabolic engineering[3]. Several interaction proteomics techniques have been developed that can detect metabolite-protein interactions[4]. For example, limited proteolysis-coupled mass spectrometry (LiP-MS) detects changes in a protein's susceptibility to digestion by proteinase K which may occur when a protein undergoes conformational change or binds to effectors. In LiP-SMap (limited proteolysis-small molecule mapping), interactions between proteins and an added metabolite are revealed by comparison of peptides released during partial digestion of proteins in the absence and presence of the metabolite. LiP-SMap previously revealed hundreds of metabolite-protein interactions in yeast and *E. coli* extracts, and in many cases, the altered peptides were near enzyme active sites[5].

An unexplored application area for interaction proteomics is the Calvin cycle, present in diverse bacteria, eukaryotic algae, and plants. Redox regulation of enzyme activity in the plant Calvin cycle and the surrounding network has been studied[6–8] and some metabolite regulation of key enzymes are incorporated into models of photosynthesis[9–11]. In contrast, post-translational regulation of the Calvin cycle in bacteria is less characterized[12]. The bacterial Calvin cycle is of biotechnological interest as cyanobacteria and chemoautotrophic bacteria have been modified to produce biochemicals from carbon dioxide using sunlight, electricity, or hydrogen as energy sources[13–16]. The Calvin cycle is susceptible to instability at branch points where intermediates are drained, and the kinetic parameters of cycle enzymes and branching enzymes are constrained[17,18]. Modulation of enzyme kinetic parameters ($K_M$, $K_i$, $k_{cat}$, Hill coefficient), such as by allosteric or competitive effectors, could affect the rate of light-mediated activation/deactivation or cycle stability. For example, it was recently shown that phosphoketolase in cyanobacteria is inhibited by ATP and that this regulation serves a biological function: phosphoketolase enhances cell fitness by depleting sugar phosphate intermediates of the Calvin cycle in the dark when ATP levels are reduced[19].

The Calvin cycle is present in all cyanobacteria and in approximately 7% of non-cyanobacterial genomes[20]. Among cyanobacteria, Calvin cycle enzyme sequences have significant homology, though potential metabolite-level regulation may be different. Comparisons of the transcriptomic response to changes in inorganic carbon supply suggest that the cyanobacterium *Synechocystis* sp. PCC 6803 responds primarily through biochemical regulation of enzyme fluxes, while *Synechococcus elongatus* PCC 7942 responds through transcription regulation of enzymes[21–23]. Recent in vitro characterization of enzymes from the oxidative pentose phosphate pathway of *Synechocystis* showed that several were unexpectedly inhibited by TCA-cycle metabolites[24,25]. In chemolithoautotrophs, the Calvin cycle is frequently acquired through horizontal gene transfer and may provide a growth advantage in environments poor in organic substrates due to improved cofactor recycling or in environments with mixed or fluctuating carbon sources[26–28]. Conservation of metabolite-level regulation in the Calvin cycle across bacterial families would imply core design principles for its operation, while differences may indicate adaptations specific to a certain microbial lifestyle or evolutionary trajectory[29].

Here, we applied the LiP-SMap technique to uncover new regulatory metabolite interactions with central carbon metabolism enzymes in four bacteria strains containing the Calvin cycle, *Synechocystis sp.* PCC 6803, *Synechococcus* PCC 7942, *Cupriavidus necator* (formerly *Ralstonia eutropha*), and *Hydrogenophaga pseudoflava*. *Synechocystis* is a model for studying photosynthesis, particularly because it can also metabolize glucose[30].

*Synechococcus* is an obligate photoautotroph and a model for the circadian rhythm[31]. *Cupriavidus necator* and *Hydrogenophaga pseudoflava* are chemoautotrophic betaproteobacteria in the order Burkholderiales and were chosen as non-photosynthetic Calvin cycle harboring bacteria[32–35]. All four strains are potential starting points for developing microbes to convert carbon dioxide into chemicals. LiP-SMap revealed species-specific interaction patterns for several tested metabolites, such as glyceraldehyde-3-phosphate (GAP), glucose-6-phosphate (G6P) and glyoxylate, which indicates that enzyme regulation by these metabolites may differ between autotrophic organisms. Complementary in vitro assays showed that GAP increases the catalytic activity of the enzyme fructose-1,6/sedoheptuolse 1,7-bisphosphatase (F/SBPase) in both *Synechocystis* and *Cupriavidus necator* F/SBPase in reducing conditions, suggesting a conserved feed-forward activation mechanism in the Calvin cycle. In contrast, G6P stimulated F/SBPase from *Cupriavidus* but not F/SBPase from *Synechocystis*.

## Results

**Assessment of LiP-SMap method and data**. The LiP-SMap protocol developed by Piazza et al. and previously applied to *E. coli* was applied to four autotrophic bacteria here, with some modifications (Fig. 1)[5]. Cells were grown photoautotrophically (*Synechocystis* and *Synechococcus*), on $H_2$ and $CO_2$ (*Hydrogenophaga*) or on formate (*Cupriavidus*) to ensure expression of the Calvin cycle enzymes. Cultures were harvested during stable exponential growth and lysed. The proteomes from cell lysates were extracted and filtered to remove endogenous metabolites, which reduced metabolite amounts by >90% and resuspended in a buffer containing 1 mM $MgCl_2$. The treatment metabolite was added to four aliquots of the proteome extract, and buffer was added to another four aliquots, which served as negative controls. Proteome extracts were then digested partially by proteinase K, followed by tryptic digestion with a mixture of the endopeptidases trypsin and LysC. Peptides were subsequently quantified using liquid chromatography-mass spectrometry. Proteins with at least one significantly altered peptide were assigned as metabolite-interacting proteins.

To assess the capability of LiP-SMap to detect changes in protein structure, we first tested the effect of reducing and oxidizing agents on *Synechocystis* cell extracts, using added dithiothreitol (DTT) and 5,5′-dithiobis-2-nitrobenzoic acid (DTNB), respectively. The addition of DTT to 1 mM resulted in altered peptides in 66 proteins, while the addition of DTNB to 50 µM altered peptides from 244 proteins, including known redox-sensitive enzymes phosphoribulokinase, F/SBPase, and Rubisco (Supplementary Fig. S1, Supplementary Data S1). A recent study identified 611 redox-sensitive proteins in *Synechocystis*[36], and 84% of the DTNB-affected proteins and 25% of the DTT-affected proteins were in this group. Since *Synechocystis* proteome extracts were sensitive to both DTT and DTNB, the proteins may be partially oxidized in the cell at the time of harvesting. However, peptide oxidation can occur at many steps throughout the proteomics workflow[37], which complicates estimates of the extent of reduction during a given cultivation condition.

Between 8000 and 15,000 peptides were detected in each LiP-SMap experiment, resulting in 5 peptides detected per protein on average (Supplementary Fig. S2, Supplementary Data S2–S5). Maximum protein counts were 1896 for *Synechocystis*, 1682 for *Synechococcus*, 2032 for *Cupriavidus*, and 1752 for *Hydrogenophaga*. The peptide coverage of Calvin cycle enzymes was generally high, averaging 14 peptides per enzyme (minimum 4, maximum 37), with a sequence coverage of approximately 50%

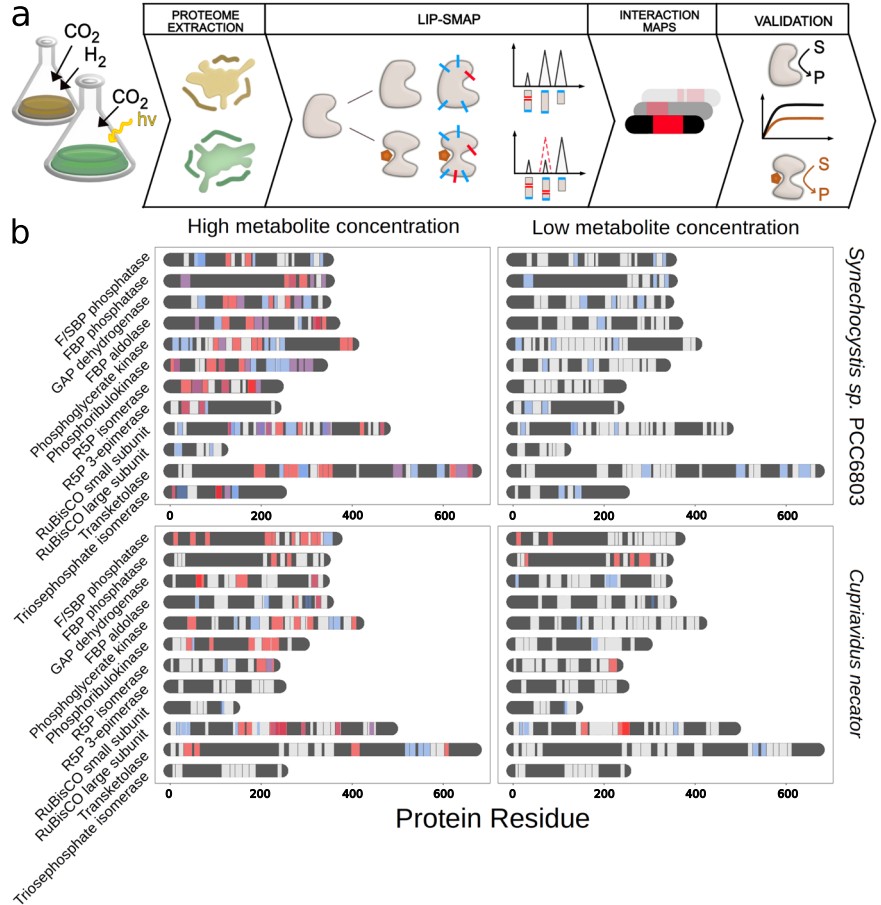

**Fig. 1 Lip-SMap on extracted proteomes of photosynthetic and chemoautotrophic bacteria. a** Workflow of Lip-SMap based on Piazza et al.[5]. Blue ticks represent digestion by trypsin/LysC, and red ticks represent digestion by proteinase K. Peptides cut by proteinase K are not detected, and differences in such digestion cause differentially abundant peptides. **b** Peptide coverage of Calvin cycle enzymes. LiP-SMap-detected peptides from Calvin cycle enzymes of *Synechocystis* and *Cupriavidus* are colored in light gray. Regions of proteins with no peptide coverage are colored in dark gray. Significantly altered peptides when extracts were treated with GAP and acetyl-CoA are colored in red or blue, respectively. Peptides that are significantly altered for both metabolites are colored in purple. For high-concentration tests, 5 mM GAP or 10 mM acetyl-CoA was added. For low-concentration tests, 0.5 mM GAP or 1 mM acetyl-CoA was added.

(Fig. 1). As expected, higher peptide coverage was observed from metabolite-interacting proteins than from non-interacting proteins (Supplementary Fig. S3). To gauge the technical reproducibility of LiP-SMap, we compared the results from two consecutive LiP-SMap experiments on the same *Synechococcus* lysate treated with 10 mM glyoxylate, a metabolite that affected a moderate number of peptides. Out of 155 peptides significantly affected ($q < 0.01$) in at least one experiment, 48 were mutual (31%). A 35–65% overlap of significant peptides among replicates has been reported previously for MS-based proteomics[38]. While the overlap in significant peptides was moderate, the log2FC of peptides was correlated across the two glyoxylate treatments, with $r = 0.88$ for significant peptides using a $q < 0.01$ cutoff, $r = 0.71$ for peptides using a $q < 0.05$ cutoff, and $r = 0.49$ for all peptides (Supplementary Fig. S4). To summarize, many peptides are affected similarly in independent metabolite treatments, but high variances in LiP-SMap data can result in their exclusion from overlap calculations.

**Interactions of metabolites with Calvin cycle and surrounding enzymes.** A panel of metabolites was screened for interactions with proteome extracts from *Synechocystis*, *Synechococcus*, *Cupriavidus*, and *Hydrogenophaga*. These bacteria use the Calvin cycle to fix $CO_2$ but differ in terms of phylogeny, energy source,

glycolytic pathways, and natural habitat. Metabolites were chosen based on their potential to act as a regulatory signal, such as metabolites located at end-points or branch-points of metabolic pathways or as representatives of energy and redox status. For each metabolite, we tested two concentrations, typically 1 mM and 10 mM (Supplementary Tables S1 and S2). The high metabolite concentrations were intended to mimic spikes that occur during environmental shifts and perturbations, which may require rapid regulation of enzyme activity[39–42]. There were more detected interactions at the high concentrations than at the low concentrations. Typically, >90% of interactions from the low-concentration treatment were also observed in the high-concentration treatment (Supplementary Fig. S5). The log2FC and significance scores for all detected peptides for each tested metabolite and bacterial strain are provided in Supplementary Data S2–S5.

To compare metabolite-protein interactions between species, we extracted a list of all proteins affected by any metabolite and grouped them according to strain and KEGG orthology groups (KOGs) (Supplementary Fig. S6, Supplementary Data S6). Principal component analysis (PCA) was used to cluster and compare metabolite-KOG interaction patterns between the species. Differences in interaction partners between species were observed at the high metabolite concentrations, as evidenced by

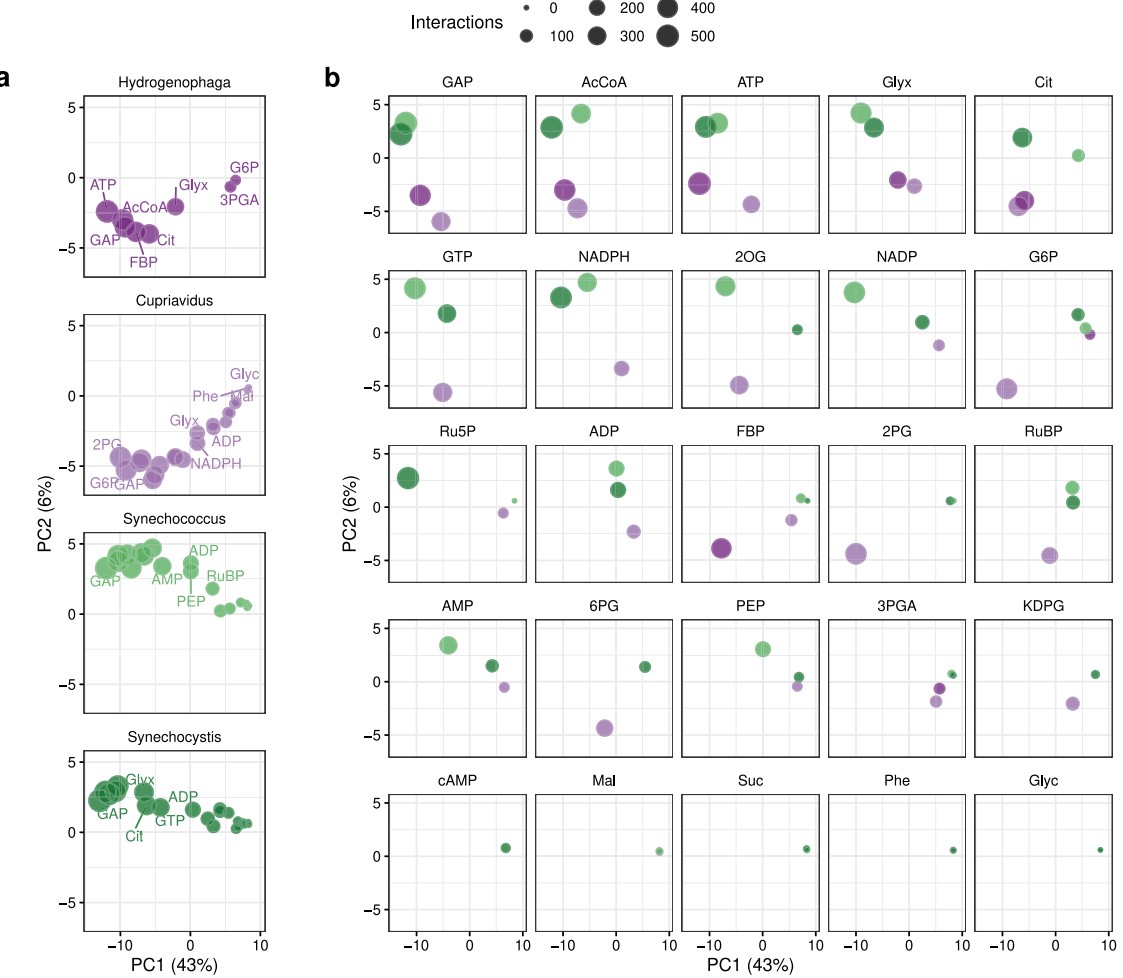

**Fig. 2 Similarity of ortholog interaction patterns, high added metabolite concentration.** Principal components were calculated from the presence or absence of interaction with each of 477 orthologs ("Methods"). All data points shown here are from the same principal component analysis but split per organism (**a**) or metabolite (**b**) to reduce overplotting. Percentages indicate the fraction of the total variance captured by the principal components.

separation among species on PCA plots for each metabolite (Fig. 2). For example, in the case of glyceraldehyde-3-phosphate (GAP) and acetyl-CoA, metabolites with more than 200 KOG interactions in all four species, interactions in the photoauto-trophs clustered apart from those in the chemoautotrophs. Glucose-6-phosphate (G6P), an entry metabolite of the pentose phosphate and the Entner-Doudoroff (ED) pathway, showed a high number of interactions in *Cupriavidus* that clustered apart from those in other species. In contrast, some metabolite-KOG interactions were similar in all species, such as metabolites in lower glycolysis and the tricarboxylic acid cycle. As fewer interactions were detected at the low metabolite concentrations, there was a weaker separation of the species in the PCA analysis (Supplementary Fig. S7).

Interactions with acetyl-CoA, ATP, citrate, GTP, and NADPH were widespread in all four microbes. These metabolites have $Mg^{2+}$ chelating properties[43,44] and may sequester $Mg^{2+}$ from proteins in extracts. Indeed, the number of ATP interactions in *Synechocystis* proteome extracts was reduced when $Mg^{2+}$ in the LiP-SMap buffer was increased from 1 mM to 3 mM, indicating that $Mg^{2+}$ chelation is a main contributor to the extensive interactions of ATP (Supplementary Fig. S8, Supplementary Data S7). Recent studies showed that citrate, as well as other metabolites of the TCA cycle, inhibit several enzymes of the pentose phosphate pathway and Calvin cycle in *Synechocystis*[24,25,45]. Comparison of LiP-SMap data showed

moderate overlap with these reported enzyme regulations. For example, of four reported inhibitors of glucose-6-phosphate dehydrogenase, two were also LiP-SMap hits (citrate and NADPH), and of two reported inhibitors of 6-phosphogluconate dehydrogenase, one was a LiP-SMap hit (citrate). In contrast, L-phenylalanine is an example metabolite that showed few but specific interactions. In *Synechocystis*, only three proteins interacted with L-phenylalanine ($q < 0.05$): two subunits of phenylalanine-tRNA ligase (PheTS) and acetolactate synthase (IlvH, *sll0065*). The enzyme 3-deoxy-D-arabino-heptu-losonate-7-phosphate-synthase (DAHPS), a key step of the shikimate pathway that is known to be allosterically inhibited by aromatic amino acids, had an interaction with L-phenylalanine near the significance cutoff (*sll0934*, $q = 0.07$)[46,47]. This short list of interactors shows the potential accuracy of LiP-SMap, even though interactors may be missed due to low peptide coverage.

Next, we examined interactions of metabolites with enzymes in the Calvin cycle and pathways that siphon carbon out of the cycle among the four bacteria (Fig. 3, Supplementary Fig. S9). Calvin cycle enzymes in these bacteria are phylogenetically diverse, though the cyanobacteria enzymes are more closely related to each other than to the chemoautotroph orthologs (Supplementary Data S8). As previously reported for *E. coli*, LiP-SMap revealed many more interactions than previously known. Some metabo-lites interacted more with the Calvin cycle enzymes of certain species. For instance, the photorespiratory intermediate

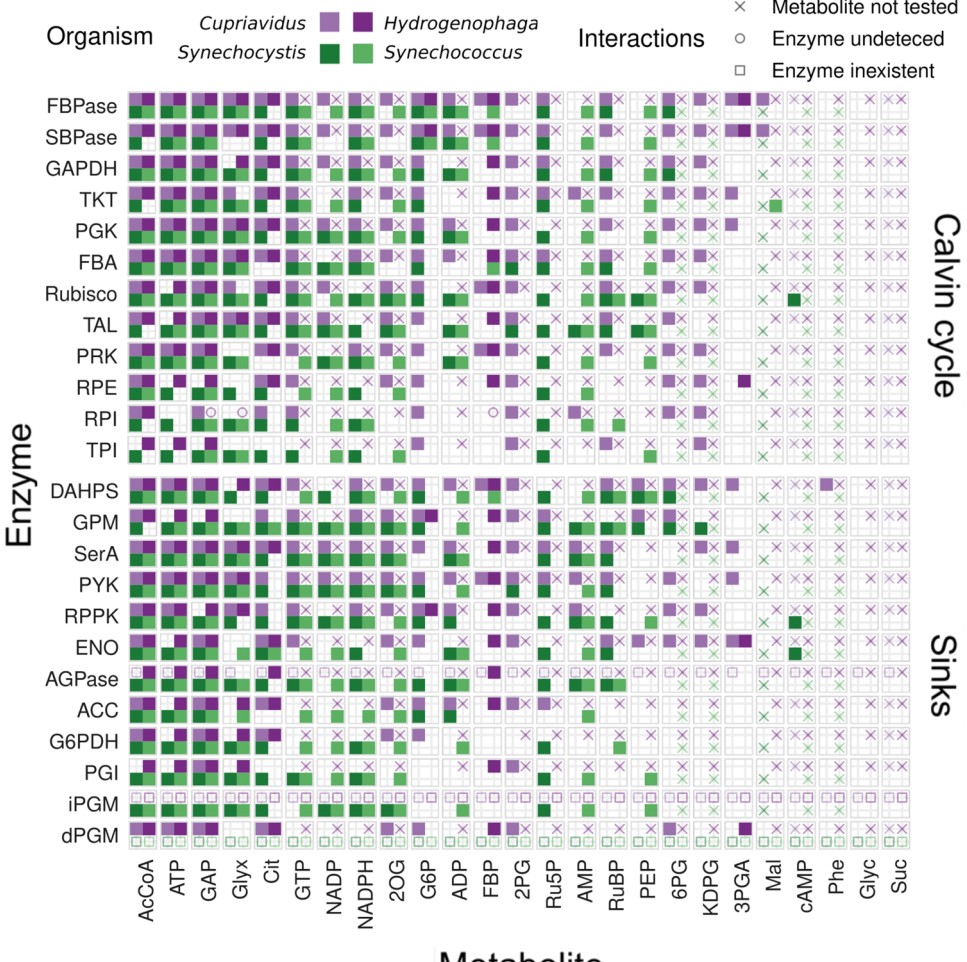

**Fig. 3 Interactions of Calvin cycle enzymes and selected central carbon metabolism enzymes with metabolites, high added metabolite concentration.** Interactions between metabolites (columns) at high concentration and enzymes (rows) identified by E.C. number are shown for each organism by tiles filled with the corresponding color. A blank tile indicates that the interaction was not detected while missing protein data is indicated by a symbol. A cross indicates that the particular metabolite was not tested, a circle indicates that the protein was not detected, and a square indicates that there was no such enzyme in the corresponding genome. See Supplementary Fig. S9 for a plot of interactions with added metabolite at low concentrations.

glyoxylate showed extensive interactions in *Synechocystis*, even at low concentrations (Supplementary Fig. S9). Calvin cycle enzymes from *Cupriavidus* were particularly sensitive to intermediates of the pentose phosphate and ED pathways. In summary, while there were some interactions observed in all species, primarily acetyl-CoA, ATP and GAP, most metabolites showed species-specific interactions with Calvin cycle enzymes. The overlap between *Synechocystis* and *Synechococcus* was only modest, despite the high homology among their enzyme sequences.

**Validation of metabolite interactions and effect on enzyme activity**. An interaction detected by LiP-SMap in a proteome extract does not necessarily mean that the metabolite interacts directly with the protein or that protein function is affected. To test if metabolite-enzyme interactions identified by LiP-SMap tended to affect the catalytic activity of enzymes, we purified F/SBPase and transketolase from *Synechocystis* and *Cupriavidus* and assayed the enzymes in vitro in the presence of selected metabolites that showed LiP-SMap interactions. F/SBPase catalyzes two irreversible steps in the bacterial Calvin cycle and was shown to have a significant effect on the rate of $CO_2$ fixation in cyanobacteria in some conditions[17,48–51]. The F/SBPase reactions are

far from equilibrium, and such enzymes are more likely to be post-translationally regulated[52]. While the transketolase step of the Calvin cycle operates reversibly and close to equilibrium[42], some studies have predicted that transketolase activity could also have significant control over $CO_2$ fixation as well as flux out of the cycle[53].

*Synechocystis* F/SBPase (syn-F/SBPase) and *Cupriavidus* F/SBPase (cn-F/SBPase) activity on the substrate fructose-1,6-bisphosphate was screened using Malachite Green (MG) detection of released $P_i$. Enzyme thermal stability was also assessed (Table 1, Supplementary Table S3). The addition of GAP, which showed interaction with the F/SBPase from all four microbes in the LiP-SMap data, stimulated both syn-F/SBPase and cn-F/SBPase activity by 50–70% (Fig. 4). This stimulation was only observed at low substrate concentrations, indicating that GAP reduces enzyme $K_M$. In contrast, the GAP isomer dihydroxyacetone phosphate did not have an effect on enzyme activity (Supplementary Fig. S10). GAP also caused a small thermal shift of both syn-F/SBPase and cn-F/SBPase (Supplementary Fig. S11), suggesting that LiP-SMap and enzyme kinetics effects are due to a conformational change mediated by GAP. The addition of NADPH inhibited enzyme activity of both syn-F/SBPase and cn-F/SBPase, though an effect on $T_m$ was only observed for

**Table 1 Effect of LiP-SMap metabolites on F/SBPase activity and thermal stability in vitro.**

| Effector (mM) | LiP-SMap interaction | Kinetics change | $T_m$ change |
|---|---|---|---|
| **syn-F/SBPase** | | | |
| Acetyl-CoA (2) | Yes | Not significant | Not significant |
| AMP (0.25) | No[a] | Inhibit | Increase |
| Citrate (5) | Yes | Inhibit | Decrease |
| GAP, +DTT (0.5) | Not tested | Stimulate | Increase |
| GAP, −DTT (0.5) | Yes | Inhibit | Not significant |
| NADPH (3) | Yes | Inhibit | Not significant |
| G6P (2) | No | Not significant | Not tested |
| **cn-F/SBPase** | | | |
| AMP (0.25) | No | Inhibit | Increase |
| GAP, +DTT (0.5) | Not tested | Stimulate | Not significant |
| GAP, −DTT (0.5) | Yes | Not tested | Not significant |
| NADPH (3) | Yes | Inhibit | Increase |
| G6P (2) | Yes | Stimulate | Not significant |

Changes in kinetic parameters were determined by enzyme kinetic assays. The kinetic effect of a metabolite was considered significant for $p > 0.05$ (comparing kinetic parameters) and a maximum change in rate >20%. Changes in melting temperature ($T_m$) of more than 2 °C were considered significant.
[a]Interaction with AMP $q = 0.056$.

cn-F/SBPase (Supplementary Figs. S12 and 13). The similar kinetic effects of NADPH and GAP on both F/SBPase enzymes suggest evolutionary convergence, as syn-F/SBPase (class II) and cn-F/SBPase (class I) have a similar monomeric fold but little sequence similarity[54,55]. In contrast to GAP and NADPH, which affected both enzymes, G6P stimulated the cn-F/SBPase up to 100% but had no effect on the syn-F/SBPase (Supplementary Fig. S14). The strong effect of G6P effect on the cn-F/SBPase is in agreement with LiP-SMap data, where G6P showed interaction with cn-F/SBPase at both high and low concentrations. The addition of AMP, a known allosteric effector of syn-F/SBPase, completely abolished the activity of the syn-F/SBPase but had a weaker effect on cn-F/SBPase.

We also compared interactions detected from LiP-SMap on proteome extracts to LiP-SMap on purified protein, using recombinant syn-F/SBPase. For the purified syn-F/SBPase, we detected approximately 40 peptides and protein coverage was >90%. The addition of GAP and NADPH resulted in 4 and 6 significantly altered peptides ($q < 0.05$), respectively, more than were found in the proteome extract LiP-Smap (2 for GAP and 3 for NADPH). Altered peptides from the purified enzyme included those altered in the proteome extract LiP-SMap. This agreement indicates that LiP-SMap hits for these metabolites in proteome extracts were not due to the enzymatic conversion of the metabolite by endogenous enzymes (Supplementary Fig. S15).

The syn-F/SBPase is redox regulated, which manifests in vitro as a stimulation of enzyme activity and a change in oligomeric state when the reducing agent DTT is added[55]. Since GAP was the only metabolite tested that stimulated syn-F/SBPase activity, we were interested in probing whether a reducing environment and GAP stimulation were synergistic. In contrast to the stimulating effect in reducing conditions, the addition of GAP reduced syn-F/SBPase activity in the absence of DTT (Supplementary Fig. S16). In the nanoDSF datasets used to calculate melting temperature, we noticed a peak shoulder representing a secondary syn-F/SBPase population when DTT was omitted from the buffer (Supplementary Fig. S16). This population disappeared with increasing DTT, and visible fractions of this shoulder correlated with a strong decrease in activity. suggesting that it may represent the non-active dimeric state of syn-F/SBPase reported previously[55]. Moreover, the fraction of this shoulder in the absence of DTT increased when pre-incubating the sample at 30 °C for different durations, which may either indicate aggregation or the dissociation of the active enzyme tetramer

into the inactive state over time. The addition of GAP during pre-incubation increased the abundance of the peak shoulder. Considering that GAP contains a reactive aldehyde group and showed interactions with many proteins in Lip-SMap, an unspecific effect, such as inducing protein aggregation in oxidative conditions, appears plausible, explaining why GAP reduces enzyme activity in the absence of DTT. The addition of GAP in the presence of DTT did not cause a peak shift in the light scattering data, indicating that the stimulating effect of GAP in the presence of DTT likely operates by a different, more specific mechanism. To exclude the possibility that GAP affected the enzyme assays by interacting with Malachite Green or other components, we also measured the enzyme reaction product fructose-6-phosphate from some syn-F/SBPase reactions via LC-MS. The observed trends, namely GAP stimulation with DTT and inhibition in the absence of DTT, were similar to those observed in the Malachite Green assays, though variation among replicates resulted in nonsignificant differences for GAP activation (Supplementary Fig. S17).

The transketolases from *Synechocystis* and *Cupriavidus* (syn-TKT and cn-TKT) were also purified and screened for enzyme activity in vitro in the presence of metabolites that showed a LiP-SMap interaction in any of the four species. The most prominent effects on transketolase kinetics were observed from added AMP and dihydroxyacetone phosphate, which specifically reduced the activity of *syn*-TKT and *cn*-TKT, respectively (Supplementary Figs. S18–20). While ATP and ADP inhibition of transketolases has been reported[56], inhibition by AMP has not. Fewer than half of the TKT-interacting metabolites detected by LiP-SMap altered TKT catalytic activity in vitro (5/13 for *syn*-TKT and 3/10 for *cn*-TKT), and only a few affected kinetic parameters by more than 20% or significantly affected melting temperature in thermal shift assays (Supplementary Table S4, Supplementary Data S9).

A benefit of LiP-SMap is that it provides peptide-level information on where metabolites interact with a protein. For syn-FBPase, both GAP and NADPH affected peptides originating from near the active site, a region distinct from the known AMP allosteric site (Fig. 5). To confirm that GAP and NADPH regulation was separate from AMP regulation, we created a single amino acid exchange variant (R194H) of a residue located in a β-sheet that connects the substrate-binding site to the AMP-binding site. This mutant lost AMP sensitivity but retained sensitivity to both GAP activation and NADPH inhibition (Supplementary Fig. S21).

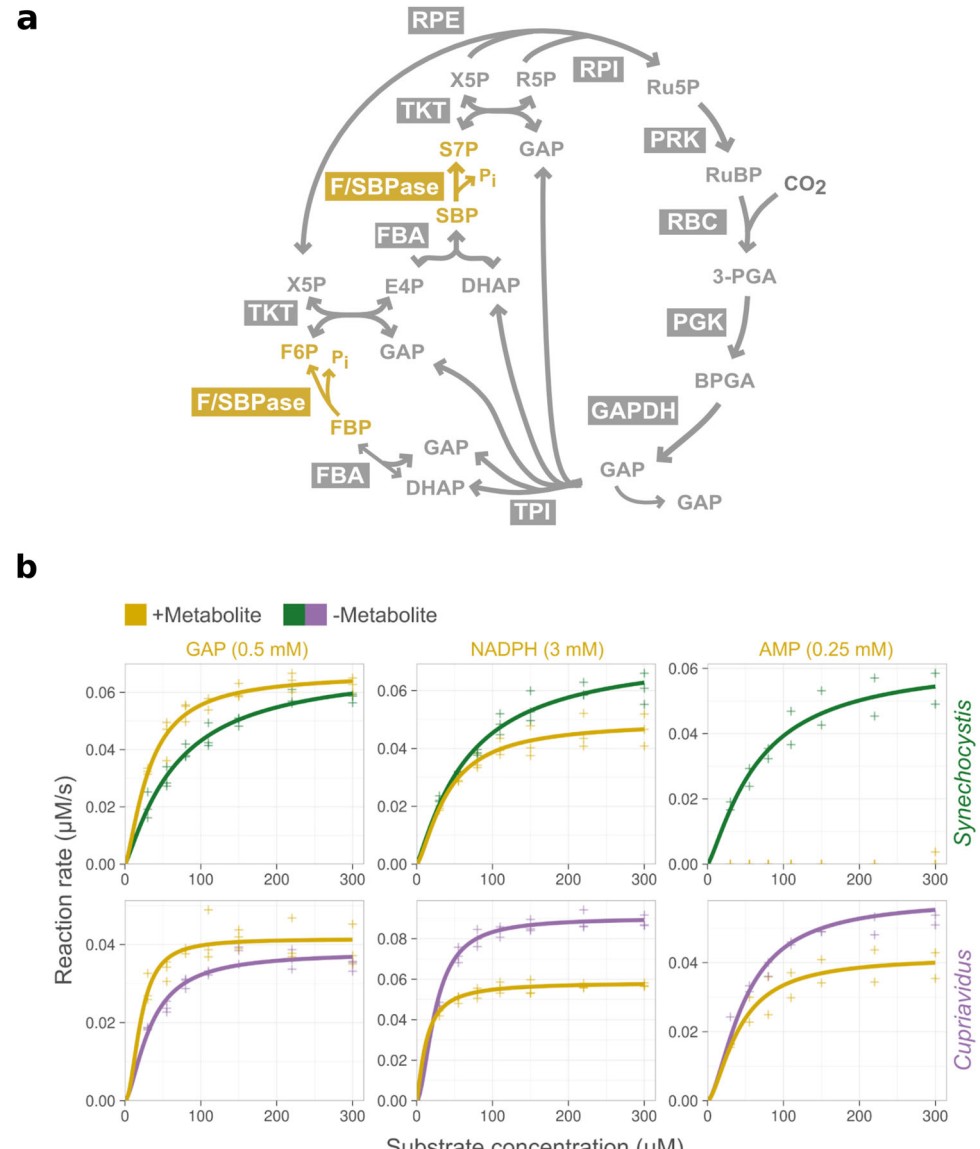

**Fig. 4 Effect of selected LiP metabolites on the activity of F/SBPase from _Synechocystis_ and _Cupriavidus_. a** Overview of bacterial Calvin cycle with reactions catalyzed by F/SBPase colored in yellow. **b** Initial rates of syn-F/SBPase and cn-F/SBPase were measured at different substrate concentrations in the presence of selected metabolites (yellow) or absence of added metabolites (green and purple for syn-F/SBPase and cn-F/SBPase, respectively). Lines represent data fit to the Hill rate equation. The concentration of added metabolite is indicated in parenthesis. See Supplementary Table S3 for kinetic data in the presence of additional metabolites.

**Predicted effects of enzyme-metabolite interactions on flux control in _Synechocystis_.** We next evaluated the effects of the regulatory interactions of GAP and NADPH on F/SBPase on flux control in the Calvin cycle and metabolic stability, using in silico ensemble modeling[17] (Fig. 6). Two Calvin cycle models were considered, a Base model with no F/SBPase regulations, and an F/SBPase model, which was the same as the Base model but with NADPH inhibition and GAP activation added to the F/SBPase rate equation. For each model variant, a large set of possible metabolic states was generated from randomly sampled metabolite concentrations and enzyme kinetic parameters ($V_{max}$, $K_M$, $K_i$, $K_a$), each satisfying the same steady-state flux distribution. The two resulting model ensembles (~3 million models each) were then assessed for system stability, which refers to the ability of the system to dynamically return to its metabolic state upon an infinitesimal small perturbation of the metabolite concentrations. The addition of regulation on F/SBPase did not alter stability

significantly, with a median stability over all parameter sets of 91% and 89% for the Base and F/SBPase models, respectively. Furthermore, the added F/SBPase regulation did not significantly alter the metabolite concentrations at which the system tends to be more or less robust.

The fully parameterized ensemble of kinetic models enables the quantification of flux control using metabolic control analysis, resulting in flux control coefficients for each reaction. In the Base model, the reactions supplying ATP and NADPH (e.g., light reactions in photosynthetic microbes) and supply of phosphate had positive FCCs over many other reactions, emphasizing their importance in autotrophic metabolism (Supplementary Figs. S22 and 23 for all FCCs). While Rubisco showed positive flux control only over reactions downstream of the Calvin cycle, F/SBPase, phosphoglycerate kinase, glyceraldehyde phosphate dehydrogenase and phosphoribulokinase had flux control over many cycle reactions. The F/SBPase model variant generated FCCs are

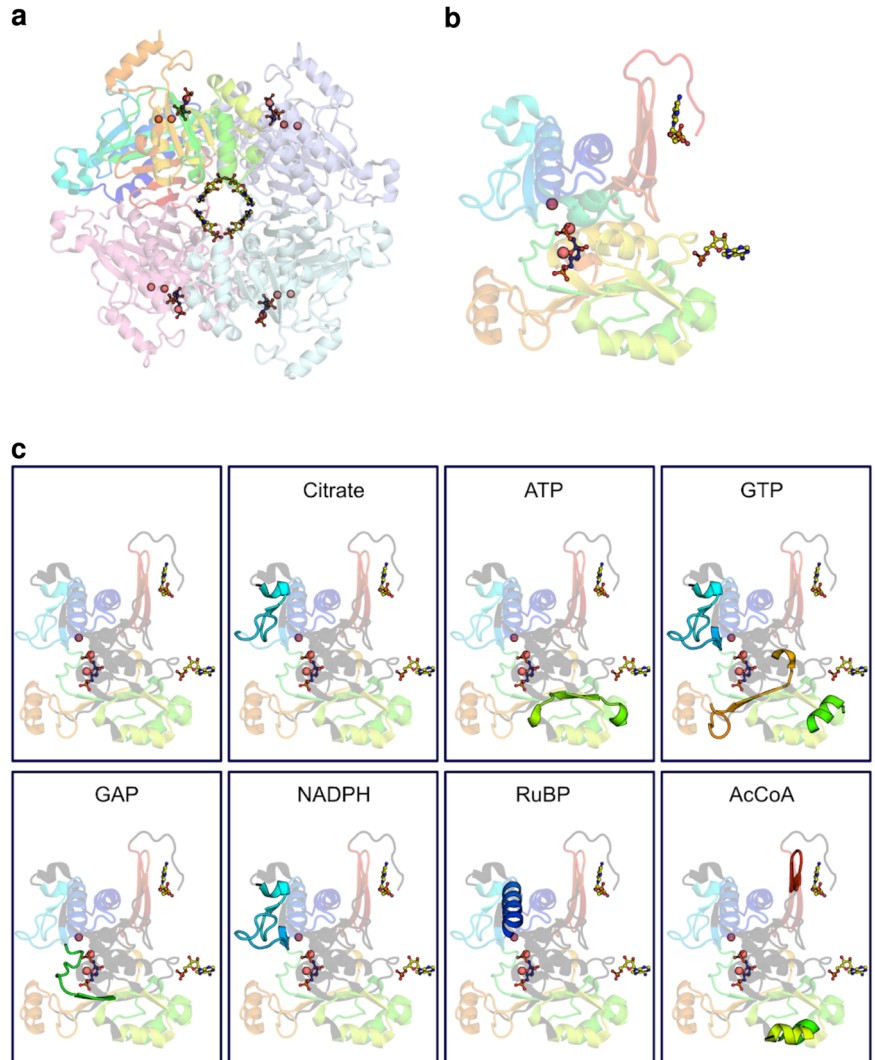

**Fig. 5 Structure of *Synechocystis* F/SBPase showing peptide coverage and affected peptides from LiP-SMap. a** Structure of F/SBPase as a homotetramer from Protein Databank reference 3RPL[55]. The substrate fructose-1,6-bisphosphate is shown in blue sticks and active site $Mg^{2+}$ ions as red spheres. The allosteric inhibitor AMP binds at the central interface of the tetramer and is shown as yellow sticks. **b** Monomer of F/SBPase colored according to different structural elements, showing interaction with fructose-1,6-bisphosphate and AMP molecules. **c** Uppler left panel shows a monomer of F/SBPase with peptides detected by LC-MS as colored ribbons. Peptides that were not detected are gray ribbons. Right panels outline which peptides were affected by the indicated metabolite.

similar to the Base model but with some distinctions. Most prominently, the flux control exerted by F/SBPase over other reactions increased, signifying higher sensitivity of these reactions to F/SBPase reaction flux. The added regulations amplify the role of F/SBPase role in controlling $CO_2$ fixation, as other reactions have reduced flux control coefficients. Flux control exerted by ATP and NADPH supply was also reduced in the F/SBPase model, rendering the system with added regulation less sensitive and thereby more stable toward potential perturbations in ATP and NADPH supply. The changes in control coefficients in the F/SBPase model are a result of both the direct effects of GAP and NADPH on F/SBPase, as well as the response of the whole system and the interactions between all participating entities.

## Discussion

The chemoproteomic workflow LiP-SMap was applied to reveal metabolite-level regulation of enzymes within the Calvin cycle and central carbon metabolism in four autotrophic bacteria. We found that some tested metabolites interacted extensively in all

organisms, such as ATP, GTP, GAP, acetyl-CoA and citrate. The extent of interactions at low added metabolite concentrations (0.5–1 mM) was significantly less than at high concentrations (5–10 mM). Metabolite control of enzyme activity is therefore more likely when metabolite levels spike, such as during environmental shifts or if synthetic metabolic pathways are installed[57]. In general, the LiP-SMap technique detects many interactions that alter proteinase K access but do not significantly alter protein conformation, as fewer than half of the LiP interacting metabolites for F/SPBase and transketolase showed an effect on thermal stability or enzyme activity.

Among the validated interactor metabolites was the Calvin cycle intermediate GAP, which interacted with many proteins in all four species. Clustering analysis showed that a subset of GAP interactions in the photoautotrophs were different compared to those in the chemoautotrophs. GAP intersects several central metabolic pathways in bacteria[58], and different utilization of these pathways between species may require specific regulation by GAP. The feed-forward activation of F/SBPase by GAP revealed here could work to prevent excessive GAP accumulation and

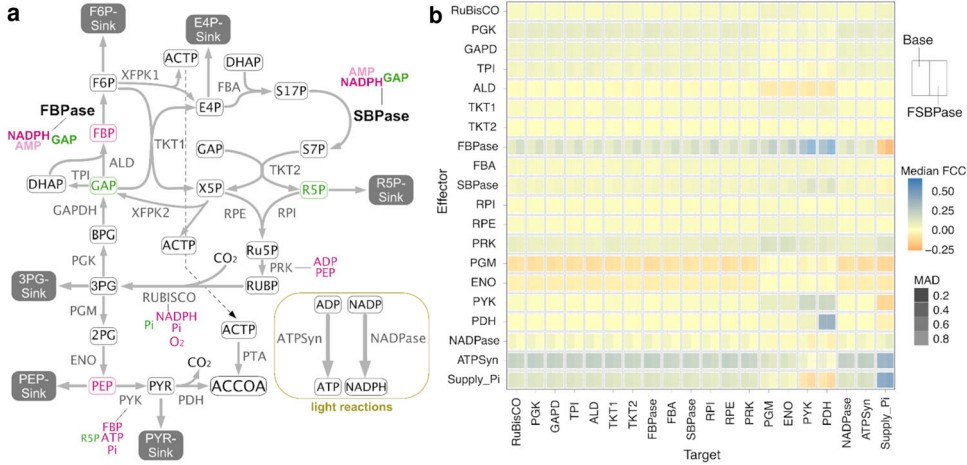

**Fig. 6 Addition of regulation on F/SBPase activity alters flux control in the *Synechocystis* Calvin cycle. a** Schematic overview of the modeled *Synechocystis* metabolic network showing all included biochemical regulations. Interactions identified from LiP-SMap and verified as modulating F/SBPase are in bold. Red text indicates inhibition of activity, and green text indicates stimulation of activity. AMP inhibition was omitted from the model. **b** Median flux control coefficients and median absolute deviation (MAD) over the entire model ensembles for both model variants. The Base model variant included no added regulation to F/SBPase, and the F/SBPase model variant included GAP and NADPH regulation.

increase Calvin cycle flux in response to up-shifts in energy or $CO_2$ levels in the growth environment. Feed-forward regulation in central metabolism is known in glycolysis and gluconeogenesis, as pyruvate kinase in eukaryotes and bacteria is stimulated by the glycolysis intermediate fructose bisphosphate[59,60] and the type I FBPase in *E. coli* is stimulated by the gluconeogenesis substrate phosphoenolpyruvate[2,61]. In photosynthetic microbes, intracellular levels of GAP, DHAP, fructose bisphosphate and sedoheptulose bisphosphate oscillate with light[40–42,62,63]. In the absence of DTT, GAP inhibits *Synechocystis* F/SBPase in a mechanism that involves aggregation into an inactive state. In the cellular context, this inactivation, in addition to disulfide formation and AMP inhibition, could work to prevent a futile cycle forming between phosphofructokinase and F/SBPase, though phosphofructokinase flux in the dark in *Synechocystis* is likely minimal[63,64].

Rapid post-translational regulation of glycogen metabolism appears to be an important feature of photoautotrophic metabolism[65,66]. ADP-glucose pyrophosphorylase, which catalyzes the first committed step in starch synthesis, interacted with more metabolites in *Synechocystis* (7 metabolites) and *Synechococcus* (7) compared to the chemoautotrophs *Cupriavidus* (1) and *Hydrogenophaga* (5). As *Cupriavidus* lacks ADP-glucose pyrophosphorylase, UDP-glucose pyrophosphorylase was compared[67]. The photorespiration metabolite glyoxylate interacted with ADP-glucose pyrophosphorylase and phosphoglucomutase in both cyanobacteria strains, which suggests that elevated levels of glyoxylate in response to inorganic carbon limitation may participate in the associated activation of glycogen degradation[68,69], though we did not test this. Phosphoglycolate, also a photorespiratory metabolite, may serve as a general carbon status indicator in cyanobacteria and trigger transcriptional expression of carbon uptake[70]. In *Synechocystis*, phosphoglycolate interacted with multiple ribosomal subunits, as well as hibernation and elongation factors, suggesting a direct effect on protein translation. Other interactions were among enzymes in nucleic acid synthesis (PyrE, PyrH, Slr1616 and Slr1619), a phosphate transporter (PtsS), and ATPase subunits AtpA and AtpB. In *Arabidopsis*, phosphoglycolate negatively affected the activities of the Calvin cycle enzymes SBPase and triosephosphate isomerase[71]. SBPase is not present in bacteria, and triosephosphate isomerase was not among the affected proteins in our

study. However, phosphoglycolate did interact with Calvin cycle enzymes transaldolase and fructose bisphosphate aldolase.

The enrichment of interactions of G6P and 2-dehydro-3-deoxy-D-gluconate-6-phosphate with *Cupriavidus* proteins may be related to the preferred usage of the ED pathway for sugar catabolism in this microbe[72], as these ED intermediates may signal sugar availability. In *Cupriavidus*, the Calvin cycle operates simultaneously and parallel to the ED pathway; G6P derived from glucose does not enter the cycle[27,28]. Stimulation of cn-F/SBPase by G6P could accelerate re-assimilation of $CO_2$ emitted during glycolysis. In *E. coli*, where EMP glycolysis and gluconeogenesis cannot operate simultaneously due to overlap, the FBPase was found to be inhibited by G6P, a regulation that effectively turns off gluconeogenesis during growth on glucose[73]. In *Synechocystis*, G6P derived from glucose enters the Calvin cycle instead via phosphoglucose isomerase, and fluxes through EMP or ED glycolysis pathways are small[74]. In this case, regulation of syn-F/SBPase activity by G6P may not be beneficial for cell fitness.

Interactions detected by LiP-SMap are not always direct but can arise from secondary effects of metabolite addition, such as $Mg^{2+}$ chelation. Metabolite chelation of $Mg^{2+}$ ions likely explains the high number of interactions observed for ATP, GTP, and citrate. The extensive interactions detected for GAP could be due to its reactivity as an aldehyde[75] or via spontaneous degradation into methylglyoxal, which reacts with lysine, arginine, and cysteine residues on proteins[76,77]. Even though such secondary effects can be confounding, they may still be relevant for metabolism in vivo. For example, excessive accumulation of citrate and ATP, which could occur during nitrogen depletion, may inactivate ribosomes and anabolic processes that depend on $Mg^{2+}$, which is particularly relevant for enzymes of photosynthesis[78,79].

Previously, inference of allosteric or metabolite-level regulation in microbial metabolism has been done through analysis of time-resolved metabolite and proteomics datasets[2,39], fitting of multi-omics steady-state data[25,80], or through coelution of proteins and metabolites from a chromatography column[81,82]. By quantifying individual peptides, the LiP-SMap method can provide insight into which area of the protein is affected by a metabolite, which provides an extra level of information compared to inference methods. However, the method is also limited by the somewhat sporadic nature of peptide detection from complex mixtures by MS; even replicate tests performed in parallel do not fully overlap

with respect to which peptides are detected[38]. As a result, even some known metabolite regulators will not be detected by LiP-SMap. The high variability in peptide coverage and in peptide quantification likely contributes to the low overlap in significant interactions among central carbon enzymes of *Synechocystis* and *Synechococcus*, even though these enzymes have high sequence homology. While PCA analysis of all interactions showed the cyanobacteria clustered together for some metabolites, the species discrimination axis had a weight of only 6%. Therefore, LiP-SMap may work best in tandem with other interaction-proteomics techniques, such as thermal proteome profiling, which relies on the quantification of proteins, not individual peptides[83]. LiP-SMap would benefit from a more accurate peptide quantification and a wider peptide coverage for improved specificity and sensitivity, respectively.

## Methods

**Cultivations and harvest**. *Cupriavidus necator* strain DSMZ 428 was grown in Ralstonia Minimal Media (RMM) with 100 mM HEPES pH 7.5 under chemostat conditions in a Photon Systems Instruments Multi-Cultivator MC-1000 OD. Each reactor tube was set up to a volume of 55 mL, $OD_{600}$ 0.05 and 3.5 g/L fructose. Once growth ceased, an inlet feed of 0.01–0.05 mL/min of 8 g/L formic acid in RMM with 100 mM HEPES pH 7.5 was initiated. Cultivations were kept running until a stable $OD_{600}$ had been observed for at least 5 doubling times.

*Hydrogenophaga pseudoflava* strain DSMZ 1084 was grown at 30 °C and 200 RPM in sealed flasks of ~135 mL containing ~25 mL DSMZ media 133 and ~110 mL of gas (70% $H_2$, 15% $CO_2$ and 15% $O_2$) at 1 bar overpressure. Cultivations were started from overnight pre-cultures grown on 1.5 g/L acetate and harvested during exponential growth at $OD_{600}$ ~ 1.0.

*Synechocystis sp*. PCC 6803 (gift from Klaas Hellingwerf, University Amsterdam) and *Synechococcus elongatus* PCC 7942 (from Pasteur Culture Collection, France) were grown in BG-11 media at 1% $CO_2$ and a light intensity of ~70 μmol/s·m² in 500 mL flasks containing 100 mL liquid until an $OD_{730}$ of ~1.0.

For each microbe, four biological replicate cultivations were performed, and immediately before harvest, the replicates were pooled. Cells were harvested by centrifugation and washed three times with cold lysis buffer before being resuspended in a small amount of lysis buffer, snap-frozen in liquid nitrogen, and stored as aliquots at −80 °C. The cyanobacteria were exposed to light at ~400 μmol·s⁻¹·m⁻² for 5 min prior to snap-freezing in liquid $N_2$.

**Proteome extraction**. Frozen aliquots were thawed on ice and lysed mechanically through bead beating by a FastPrep-24 5G lysis machine over six cycles of 45 s at 6.5 m/s with 30 s on ice between cycles. The lysate was spun down, and the supernatant was run through a Zeba Spin Desalting Column (size exclusion chromatography). Protein concentration in the desalted lysate was evaluated using a Bradford assay. The samples were kept at 4 °C throughout the procedure.

**Limited proteolysis**. For every experiment, three sample groups were created, one with no added metabolite and two with different concentrations of metabolite specified in Supplementary Table S1. Each sample group was prepared as four technical replicates with 1 μg/μL extracted protein. For limited proteolysis on purified syn-F/SBPase, the purified enzyme was reconstituted to a final concentration of 0.1 μg/uL in lysis buffer. Proteinase K was simultaneously added to all samples at a 1:100 protease to protein ratio and incubated at 25 °C for exactly 10 min before immediate denaturation. All sample groups originated from the

same cell extract, were treated in parallel with the same reagent aliquots and run on the LC-MS on the same plate.

**Complete digestion**. The protein mix was incubated at 96 °C for 3 min prior to treatment with 5% sodium deoxycholate and 10 mM DTT and another 10 min at 96 °C after. The samples were then alkylated by 10 mM iodoacetamide at RT for 30 min in the dark, after which proteases LysC and trypsin were applied at a 1:100 protease to protein ratio and incubated at 37 °C and 400 RPM in a thermocycler for 3 and 16 h, respectively. Digestion was halted by the addition of formic acid to reduce pH below 2, which caused sodium deoxycholate to precipitate. Samples were then centrifuged at $14,000 \times g$ for 10 min, after which the supernatant was removed and stored at −20 °C.

**Peptide purification**. Pipette tips packed with six layers of C18 matrix discs (20–200 μL; Empore SPE Discs) were activated with acetonitrile and equilibrated with 0.1% formic acid prior to being loaded with thawed peptide mixes. The matrix was then washed twice with one loading volume of 0.1% formic acid before being eluted with a mixture of 4:1 ratio of acetonitrile to 0.1% formic acid. The eluate was stored at −20 °C until analysis by LC-MS.

**LC-MS analysis**. Analysis was performed on a Q-exactive HF Hybrid Quadrupole-Orbitrap Mass Spectrometer coupled with an UltiMate 3000 RSLCnano System with an EASY-Spray ion source. In this, 2 μL of each sample was loaded onto a C18 Acclaim PepMap 100 trap column (75 μm × 2 cm, 3 μm, 100 Å) with a flow rate of 7 μL per min, using 3% acetonitrile, 0.1% formic acid and 96.9% water as solvent. The samples were then separated on ES802 EASY-Spray PepMap RSLC C18 Column (75 μm × 25 cm, 2 μm, 100 Å) with a flow rate of 3.6 μL per minute for 40 min using a linear gradient from 1% to 32% with 95% acetonitrile, 0.1% formic acid and 4.9% water as secondary solvent.

For proteome samples, MS analysis was performed using one full scan (resolution 30,000 at 200 m/z, mass range 300–1200 m/z) followed by 30 MS2 DIA scans (resolution 30,000 at 200 m/z, mass range 350–1000 m/z) with an isolation window of 10 m/z. The maximum injection times for the MS1 and MS2 were 105 ms and 55 ms, respectively, and the automatic gain control was set to $3 \cdot 10^6$ and $1 \cdot 10^6$, respectively. For purified protein samples, the full scan (resolution 60,000 at 200 m/z, mass range 300–1200 m/z) was followed by 10 MS2 DDA scans for the 10 most abundant peptides (resolution 60,000 at 200 m/z with an isolation window of 2 m/z). For MS1, the maximum injection time was set to 205 ms and the automatic gain control to $1 \cdot 10^6$. For MS2, the settings were 105 ms and $2 \cdot 10^5$, respectively. Precursor ion fragmentation was performed with high-energy collision-induced dissociation at an NCE of 26 for all samples.

Prosit intensity prediction model "Prosit_2020_intensity_hcd" was used to generate a predicted peptide library from a FASTA file of UniProt proteome sets (*Cupriavidus necator*: UP000008210, *Synechocystis sp*. PCC 6803: UP000001425, *Synechococcus elongatus sp. PCC 7942*: UP000002717, *Hydrogenophaga pseudoflava*: UP000293912). The data was then searched using the EncyclopeDIA version 1.2.2 search engine. Spectra for purified proteins were deconvoluted using MaxQuant v. 2.0.3.0, using Oxidation (M) and Acetyl (Protein N-term) as variable modifications and Carbamidomethyl (C) as a fixed modification. A maximum of two missed cleavages were allowed, and the false discovery rate was set to 1%.

**Data analysis**. Peptides detected in at least three replicates in every sample group were tested for differential peptide abundance

using the MSstats package (version 4.4.1) in R (version 4.3.1.). For every peptide in each metabolite concentration comparison MSstats estimated fold changes and *p*-values adjusted for multiple hypothesis testing (Benjamini-Hochberg method) with a significance threshold of 0.01. A protein was considered to interact with a metabolite supplied at low or high concentration if at least one peptide showed significant change.

*Ortholog annotations*. In order to compare metabolite-protein interaction patterns between organisms, it was necessary to determine orthologous genes. Ortholog labels from the eggNOG database were downloaded from UniProt (https://www.uniprot.org/) on 14 June 2021 for each protein in the four organisms. Version 5.0 of eggNOG was used except for proteins Q31NB2 (ENOG4108VFZ), Q31RK3 (ENOG4105KVS), and Q31RK2 (ENOG4105HKE) in *Synechococcus*, which were annotated with eggNOG version 4.1. Only the 481 orthologs found in all organisms were considered. The number of interacting proteins was counted for each ortholog and metabolite concentration in each organism. Furthermore, ortholog counts were summarized into the 20 functional categories, each represented by a single letter, e.g., "A" for "RNA processing and modification."

*Principal component analysis of interactions with orthologs*. The metabolite-protein interaction patterns of orthologs were compared between metabolites and organisms using R. The interaction per ortholog was first classified binarily so that the interaction was 1 (one) if there was at least one interaction for the ortholog in a particular combination of organism, metabolite, and concentration. Otherwise, the interaction was classified as 0 (zero). Orthologs without interactions were filtered out. A matrix with rows representing organism and metabolite, and columns containing the binary interaction classification of each ortholog, was subjected to principal component analysis (PCA; function *prcomp*). The first two principal components were then plotted in order to visualize how similar different organisms and metabolites were in terms of interaction with the full set of orthologous genes. The PCA was performed separately for low and high metabolite concentrations.

*Clustered heatmap of interactions with orthologs*. The metabolite-protein interaction patterns of orthologs, summarized per ortholog functional category, were further inspected through visualization with a heatmap with clustered rows and columns. The ortholog interaction counts were normalized to indicate the fraction of interacting orthologs within each combination of functional category, organism, metabolite, and concentration. These fractions were then used to calculate Euclidean distance (function *vegdist* from library *vegan*) followed by clustering (*ward.D2* method in function *hclust*), which determined the order of functional categories (heatmap rows), and metabolites and concentrations (heatmap columns). Organisms contributed both to row and column clustering. Finally, the ortholog interaction fractions were plotted as heatmaps, using row and column orders as described, with dendrograms clarifying the clustering (function *ggtree* from library *ggtree*).

*Phylogenetic analysis*. Sequences for Calvin cycle KEGG orthologs (KO) in module M00165, supplemented with transaldolase (K00616 and K13810), triose-phosphate isomerase (K01803), and ribulose-phosphate epimerase (K01783), were downloaded from UniProt on 14 October 2021. Each set of KO sequences was reduced in number with cd-hit version 4.8.1[84,85] by selecting the highest percent identity setting between 50% (-c 0.5) and 100% (-c 1) in 5% steps, which resulted in fewer than 1000 representative sequences. For each KO set, we added any missing

corresponding protein sequences in the four organisms studied here. Sequences were aligned using mafft version 7.453 at default settings[86]. The alignments were then used to construct phylogenetic trees with FastTree version 2.1.11 Double precision at default settings[87]. NCBI taxonomy data downloaded on 8 October 2021 was used to identify organism groups. Trees were plotted using *phytools* and *ggtree* in R in order to visualize the phylogenetic distribution of sequences and metabolite interactions for the four organisms under study.

**Cloning and transformation**. The *glpX* (*slr2094*) gene from *Synechocystis sp*. PCC 6803 and the *fpb3* (cbbFp) gene from *C. necator* were codon optimized for expression in *E. coli* and synthesized by Twist Biosciences. The genes were cloned into pET-28a(+) using Gibson assembly. The products were verified by sequencing and transformed into *E. coli* BL21 by heat shock.

The *tktA* gene from *Synechocystis sp*. PCC 6803 and the *cbbTP* gene from *C. necator* were PCR amplified from the isolated genomes using the primer pairs tktAF+tktAR and cbbTpF+cbbTpR, respectively. The backbone pET-28a(+) was linearized using the primer pair pETF+peTR after which the constructs were assembled through Gibson assembly. The products were verified by sequencing and transformed into *E. coli* BL21 by heat shock.
tktAF: 5′-CCATTTGCTGTCCACCAGACAGTGAGGAGTTT TAAGCTTGG-3′
tktAR: 5′-CCGCGCGGCAGCCATATGAACATTATGGTCG TTGCTACCC-3′
cbbTpF: 5′-CCATTTGCTGTCCACCAGATCAAGCGTCCTC CAGCAG-3′
cbbTpR: 5′-CCGCGCGGCAGCCATATGGAGATGAACGCA CCCGAACG-3′
pETF: 5′-CATATGGCTGCCGCGCGG-3′
pETR: 5′-CTGGTGGACAGCAAATGGGTCG-3′

**Production and purification of recombinant F/SBPase and TKT enzymes**. The recombinant *E. coli* BL21 strains were cultivated in 2YT media at 37 °C and 200 RPM until OD 0.4–0.6, after which overexpression of the enzyme from the pET-28a(+) plasmid was induced by the addition IPTG to 0.5 mM. The *E. coli* strains harboring the syn-*tktA*, syn-glpX, and cn-fbp3 were incubated at 37 °C for 8 h after induction, while *E. coli* strain harboring the cn-*cbbTP* gene was incubated at 18 °C for 24 h after induction. Cells were then harvested by centrifugation at 4 °C, and cell pellets were stored at −20 °C. Frozen pellets were resuspended in 3–5 mL of B-PER™ Complete Bacterial Protein Extraction Reagent (ThermoFisher Scientific) and incubated on a rocking table for ~30 min before centrifugation at $4000 \times g$. The soluble fraction was loaded onto a HisTrap Fast Flow Cytiva column (1 mL) using an ÄKTA start protein purification system and washed with 15 column volumes of wash buffer (50 mM Tris-HCl, 500 mM NaCl, 20 mM imidazole, pH 7.5) prior to elution with a stepwise gradient of elution buffer (50 mM Tris-HCl, 500 mM NaCl, 300 mM imidazole, pH 7.5). Fractions containing protein were combined, and the buffer was exchanged to storage buffer 50 mM Tris-HCl, pH 7.5 (TKT), pH 8.0 (F/SBPase) using a PD-10 Cytiva desalting column. The purified protein was quantified by Bradford assay and stored at −80 °C in aliquots.

**In vitro kinetic assays of transketolase with metabolite effectors**. Transketolase was characterized following a previously published protocol[88]. The conversion of D-ribose-5-phosphate and L-erythrulose to sedoheptulose-7-phosphate and glycolaldehyde was measured through the consumption of NADH by alcohol dehydrogenase when reducing glycolaldehyde to ethylene glycol. Relative comparisons of enzyme kinetics were made as

calculated from 8 different substrate concentrations (0 mM, 0.1 mM, 0.2 mM, 0.5 mM, 0.75 mM, 1 mM, 2 mM and 4 mM) with and without 1 mM added metabolite. The tested metabolites were 2-oxoglutarate, 2-phosphogluconate, ATP, AMP, G6P, citrate, glyoxylate, malate, NADP and dihydroxyacetone phosphate (Supplementary Data S9). The reaction mix contained 100 mM glycylglycine buffer pH 7.5, 5 mM MgCl, 2 mM thiamine pyrophosphate, 0.5 mM NADH, 100 mM L-erythrulose, 10 U alcohol dehydrogenase, 2.8 µg/mL transketolase, and D-ribose-5-phosphate to a final volume of 100 µL. Absorption was measured at 340 nm twice per minute over 30 min, starting immediately after the addition of D-ribose-5-phosphate.

**In vitro kinetic assays of F/SBPase with metabolite effectors**. In vitro enzyme activity assays were conducted to validate the kinetic effect of F/SBPase metabolite interactions detected by LiP-SMap. To determine metabolite-induced changes in enzyme kinetic parameters, initial rates were measured at eight different substrate concentrations (0, 30, 55, 80, 110, 150, 220, 300 µM) in the presence and absence of a metabolite (+M and -M). Tested metabolites were GAP, NADPH, AMP, acetyl-CoA, and citrate at 0.5 mM, 3 mM, 0.25 mM, 2 mM, and 5 mM, respectively (Supplementary Data S9). The conversion rate of fructose-1-6-bisphosphate to fructose-6-phosphate was determined from the accumulation of inorganic phosphate over time, using a Malachite Green (MG) assay adapted from a previously published protocol[89]. MG dye stock (1.55 g/L Malachite Green oxalate salt, 3 M $H_2SO_4$) was used to prepare a fresh phosphate colorimetric development solution prior to each experiment (400 µL MG dye stock, 125 µL ammonium molybdate (60 mM), 10 µL Tween-20 (11% v/v)). The development solution was filtered through a 0.2 µm syringe filter and kept in the dark. Development plates were prepared by mixing 36 µL development solution with 100 µL reaction buffer (50 mM Tris-Hcl, 15 mM MgCl₂, 10 mM DTT) lacking DTT. Enzyme solutions for +M and -M conditions were prepared in separate 8-tube PCR strips (VWR #732-1521 or low-protein binding) by mixing 25 µL reaction buffer (+M/-M) with 25 µL purified enzyme constituted in -M reaction buffer. The two strips were pre-incubated at 30 °C for 12 min in a thermocycler together with two additional PCR strips, which contained substrate at eight different concentrations in -M reaction buffer. Reactions were initiated by quickly mixing 50 µL substrate with the enzyme mixture in one of the reaction strips using a multi-pipette ([F/SBPase]$_{Final}$ = 0.42 ng/µL). A sample of 20 µL was immediately transferred to a development plate before incubating the reaction strip at 30 °C, which quenches the reaction. The initiation procedure was repeated for the second reaction strip with a 2-min delay. Samples were collected after 10, 20, and 30 min. Each sampling event was followed by an addition of 7.5 µL sodium citrate (34% w/v) to stabilize the color of the development solution. Triplicate series of phosphate standards (0–100 µM) was added to the development plate as a reference. The plate was incubated for 20 min in the dark before measuring the absorbance at 620 nm in a plate reader. The experiment was replicated at least twice. To quantify the amount of phosphate, the background absorbance measured at time zero was first subtracted from raw absorbance measurements. Phosphate standard series were then used to convert absorbances to phosphate concentrations. Outliers and phosphate concentrations that were lower than 10 µM (sensitivity threshold) or that exceeded 60% substrate conversions (10-min time points were always kept) were removed. Reaction rates were calculated as the change in phosphate concentration over time using linear regression.

In addition, one experiment was conducted where 75 µL reaction mixture (endpoint measurement rather than rate measurement) was transferred to an equal volume of methanol and analyzed on a TSQ Altis Triple Quad mass spectrometer coupled to a Vanquish UHPLC with a HESI ion source. Then, 10 µL of the sample was loaded onto an Accucore-150-amide-HILIC column (50 mm × 2.1 mm, 2.6 µm) with a flow rate of 0.4 mL per min. The samples were separated for 5 min using a linear gradient from 90% to 0% acetonitrile with 10 mM ammonium carbonate and 0.2% ammonium hydroxide in water as a secondary solvent. The mass spectrometer was run in negative mode with a voltage of 2500 V and searched for the transitions indicated in Supplementary Table S5.

**Thermal shift assay and scattering experiments**. Thermal unfolding of proteins was measured in the absence and presence of metabolites by nano differential scanning fluorimetry (F350/F330) using a Prometheus NT.48 (NanoTemper) at 95% excitation power over a temperature gradient from 20 °C to 95 °C at an increase of 1 °C per minute. Transketolase samples were prepared in 50 mM Tris-HCl pH 7.5 with 5 mM MgCl₂, 2 mM TPP, 200 ng/µL enzyme and 1 mM of metabolite. In addition, samples with and without 2 mM TPP and 10 mM DTT were also run to assay the effect of the cofactor and reductive power on protein stability. F/SBPase samples were prepared in 50 mM Tris-HCl pH 8 with 200 ng/µL enzyme, 15 mM MgCl₂, 10 mM DTT and varying concentrations of metabolite (Table 1). The effect of acetyl-CoA, GAP and citrate (2, 0.5, and 5 mM, respectively) on syn-F/SBPase was analyzed at different MgCl₂ concentrations to test whether T$_m$ changes were caused by magnesium chelation. T$_m$ changes greater than 2 °C were considered significant.

Scattering data were recorded simultaneously with fluorescence measurements for each dataset. For the analysis of protein states, the first derivative of scattering data was used. syn-F/SBPase samples (final assay concentration of 150 ng/uL) were prepared as described above in the absence and presence of 10 mM DTT and 0.5 mM GAP, respectively. The protein was pre-incubated with the respective buffers for different durations, as indicated. All pre-incubated samples were analyzed in the same run.

**Kinetic metabolic model**
*Model structure*. The kinetic model for *Synechocystis* central carbon metabolism was based on a previous model[17]. The final model contained 29 reactions connecting 36 metabolites (22 internal). Sink reactions were formulated as irreversible Michaelis-Menten-type equations. Phosphate supply followed mass action kinetics. Two model variants were created: One base model, including only the regulatory interactions in the previous version[17], and one model with interactions on F/SBPase (GAP and NADPH).

*Metabolite concentrations and flux distribution*. Due to the uncertainty associated with published metabolomics datasets, potential thermodynamically feasible metabolite concentrations describing the metabolic state were randomly sampled as performed previously[17]. Metabolite concentration ranges identified via NET analysis[29] were used as constraints for the sampling, resulting in ~3000 feasible metabolite concentration sets covering the entire thermodynamically allowable solution space. The steady-state flux distribution was obtained using a genome-scale metabolic model of *Synechocystis* as described in Janasch et al.[17]. All flux simulations were performed in Matlab R2020b using the *Gurobi Optimizer* version 9.1.1. Maximizing autotrophic growth was set as the objective function. Fluxes were manually curated to adjust the genome-scale flux distribution to the small-scale kinetic model structure and transformed into mM/min by multiplying with the cellular density of 434.78 g/L for *E. coli*[90].

*Parameter sampling.* Rate equations were generally parameterized around the corresponding metabolite concentrations by sampling the range of 0.1x to 10x metabolite concentration in logarithmic space for $K_M$ values. Inhibition constants $K_i$ and activation constant $K_a$ for the regulations identified by LiP-SMap were sampled in a narrower range of 0.5x to 2x around the metabolite concentrations used for the enzyme assays. For the activation of F/SBPase by GAP, $K_m$ could maximally be reduced by 75%. Hill coefficients for F/SBPase were sampled between 1 and 1.5. $V_{max}$ values were calculated back from metabolite concentrations, sampled kinetic constants and the steady-state flux distribution. For each of the 3151 fMCSs, 1000 parameter sets were generated, resulting in an ensemble of ~3 million kinetic steady-state models to be analyzed for stability and metabolic control.

*Metabolic control analysis.* The dynamic behavior of the models was analyzed by linearizing them around their steady state as performed previously by forming the Jacobian matrix y[17,91,92]. The stability of each model in the ensemble was evaluated by calculating the eigenvalues of the Jacobian matrix, where positive eigenvalues cause instability. Flux control coefficients were calculated for all stable parameter sets based on elasticities and concentration control coefficients as described previously[17]. The models and all code required to perform the kinetic modeling analysis are available at https://github.com/MJanasch/KX_Kinetics.

**Statistics and reproducibility.** Interaction proteomics was done in technical quadruplicates. For each experiment, plots visualizing principal component analysis, quantile-quantile analysis, *p*-value distribution, intensity distribution and peptide count per sample were generated and inspected to ensure quality of data. Obvious outliers were removed, and experiments with very abnormal statistics were re-done. Transketolase and F/SBPase assays were done in technical triplicates, and kinetic parameters were calculated by fitting reaction rates and substrate concentrations to the Hill equation using non-linear regression. A parameter change was considered statistically significant if $p < 0.05$ (Student's *t*-test, two-sided) and the effect size (difference of the mean) exceeded 20%. Obvious outliers were removed. Light scattering assays and melting temperature assays were done in technical triplicates. Obvious outliers were removed.

**Reporting summary.** Further information on research design is available in the Nature Portfolio Reporting Summary linked to this article.

## Data availability

The mass spectrometry proteomics data have been deposited to the ProteomeXchange Consortium via the PRIDE[93] partner repository with the dataset identifier PXD044412. Source data for all graphs and plots in the article can be found in Supplementary Data S9. Supplementary Data S1–S9 are available at Figshare (https://figshare.com/articles/dataset/Supplemental_Datasets/23939604). General data and quality assessment statistics, visualizations and ortholog analysis were generated by the pipeline available at https://github.com/Asplund-Samuelsson/lipsmap, implemented in R version 4.1.1 with Tidyverse version 1.3.1.

## Code availability

The code for initial proteomics analysis, transketolase kinetic assays, F/SBPase assays, melting temperature assays and light scattering assays are available at https://github.com/emilsporre/lipsmap-comp, implemented in R version 4.3.1 with Tidyverse version 1.3.2. The models and all code required to perform the kinetic modeling analysis are available at https://github.com/MJanasch/KX_Kinetics. The code used for the meta-analysis of LiP-SMap data is available at https://github.com/Asplund-Samuelsson/lipsmap.

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

## Acknowledgements

We are grateful to Michael Jahn (KTH Stockholm) for the helpful discussion on proteomics and cultivation of *Cupriavidus necator*. We thank Ralf Steuer (Humboldt University, Berlin) for discussions on kinetic modeling. Funding for this work was from the Novo Nordisk Foundation (grant numbers NNF19OC0057652 and NNF20OC0061469), the Swedish Research Council Vetenskapsrådet (grant number 2016-06160), and the Swedish Foundation for Strategic Research SSF (ARC19-0051).

## Author contributions

Conceptualization: J.K., E.S., P.S., F.E., I.P., E.P.H. Experimental proteomics: J.K., E.S., A.K., D.K., L.S., F.E. Proteomics data analysis: E.S., J.K., J.A.S., L.S., F.E. Enzyme kinetics and melting analyses: J.K., K.S., E.S. Metabolic modeling: M.J., L.Z. Writing initial draft: J.K., E.S., J.A.S., M.J., K.S., E.P.H. Editing of final draft: J.K., E.S., J.A.S., A.K., M.J., K.S., E.P.H. Funding acquisition: P.S., E.P.H.

## Funding

## Competing interests

The authors declare no competing interests.
