## [Peer Review File · Communications Biology]

Reviewers' comments:

Reviewer #1 (Remarks to the Author):

This paper uses the technique limited proteolysis small molecule mapping (LiP-SMap) to explore the role of metabolite protein interactions in the Calvin cycle and to test the impact of these on the regulation of flux. The main findings of this work is that the regulation exerted by metabolites on the function of the Calvin cycle are more numerous than previously thought. The enzyme F/SBPase was shown to be differentially regulated by GAP dependent on the redox conditions.

The work presented in the paper is novel and will wide interest to the community outside of bacterial photosynthesis. The work presented is rigorous and the conclusions drawn are well balanced.

I think this paper will influence thinking in the field and forms a proof of concept piece of work for others to follow.

Overall this is a well crafted manuscript, that is written with clarity and presents areas for further work.

Reviewer #2 (Remarks to the Author):

Sporre et al. performed a novel analysis based on proteolysis-coupled mass spectrometry to examine metabolite-protein interaction in cyanobacteria. Using four autotrophs, they performed LiP-SMap analysis and detected 8,000-15,000 peptides including those derived from Calvin cycle enzymes. They compared the peptide profiles in the presence of various chemicals and metabolites. The results showed the interaction between the Calvin cycle enzyme and metabolites such as glyoxylate, sugar phosphates, 2-OG, ATP, and so on. Finally, the authors validated the results by biochemistry determining kinetic parameters. Overall, this study is intriguing, and the novel methods contribute to the development of the understanding of cyanobacterial metabolisms. I have several comments before publication.

Major points,

To validate the results of LiP-SMap analysis, the published biochemical data can be useful, but less cited in this paper. For example, citrate is a key metabolite regulating the enzymes of the oxidative pentose phosphate pathway, but not the TCA cycle (Ito and Osanai, 2020 Biochem J.). Also, various enzymes of primary carbon metabolism (Calvin cycle, OPP pathway, glycolysis, and TCA cycle) are known to be regulated by metabolites. Please cite the biochemical papers to validate the current results.

Line 150, how do the authors determine the culture conditions for LiP-SMap? The culture conditions are important to compare the data and protein profiles are varied by cultivation conditions and growth phases.

Minor points,

Line 28, sp. (non-italic and period)

Line 132, I cannot understand why the authors used *Cupriavidus necator* and *Hydrogenophaga pseudoflava* to compare with cyanobacteria. An additional explanation is required in the Introduction.

Line 175, readers are difficult to know why the authors tested the glyoxylate-treated cells for LiP-SMap analysis (I know photorespiration is mentioned in the Results, but until that, the readers cannot

find it).

Line 220, please check the sl number for the gene encoding the cAMP receptor protein. Generally, sl numbers are four digits.

Reviewer #3 (Remarks to the Author):

Review on MS COMMSBIO-23-0305: "Metabolite interactions in the bacterial Calvin cycle and implications for flux regulation", by Professor Hudson and colleagues

Modern high throughput methods made a great contribution to our understanding of biological phenomena on the systems level. In addition to changed transcript or protein abundances, the biochemical regulation of enzyme activities is a layer significantly contributing to flux alterations under different environmental conditions. Recently, the LiP-SMap method was published that permits a high throughput analysis of protein/metabolite interactions. The method has not been applied to the Calvin cycle, which was done here for the first time. CO₂ fixation through Calvin cycle is the main driver of the global carbon cycle. Moreover, microbial CO₂ fixation is the base for the future sustainable biotechnology, because it can use the greenhouse gas for multiple applications. The authors analyzed the potential regulation of Calvin cycle enzymes and others associated with this cycle in four different bacteria. Two cyanobacterial model strains were used, which are potentially interesting as so-called green cell factories and can also serve as model for the plant Calvin cycle. In addition, two heterotrophic bacteria were analyzed, which have the potential in biotechnology to produce organic compounds from CO₂ and H₂. In the beginning, the authors provided many data, which were obtained to evaluate the methods for its application to the specific question. As usual, the study discovered a great number of potential metabolite/enzyme interactions. The authors found that the Calvin cycle regulation due to different metabolites seems to be different between autotrophic and heterotrophic bacteria. In addition to the high throughput screening, some of the observed interactions were verified by subsequent detailed enzyme assays to characterize the biochemical and structural features. Finally, the impact of the newly discovered metabolic regulation of FBP/SBPase in *Synechocystis* was embedded in a model showing its impact on the overall Calvin cycle regulation.

Points to consider:

Line 167: please discuss if the partial oxidized state of the *Synechocystis* is rather because of oxidation through the protein extraction in the presence of oxygen or might reflecting the in vivo state of proteins!

Line 171: please mention the overall coverage of the proteome of the 4 bacteria

Line 178: I guess that this number relates to overall peptide detection (which is surprisingly low), it might be better to compare the values here to the reproducibility of detected metabolite/peptide interaction LiP_SMap experiments in *E. coli* or yeast

Line 194: The data in Fig. S5 seem to differ from the data shown in Fig. 1B. A visual comparison of red or blue peptide marks in high versus low concentrations indicates a much lower interaction with GAP but also AcCoA between high and low concentration, please clarify this

Line 212: does this mean that the more specific interactions, which can be assumed to remain visible when adding lower metabolite concentrations, are more conserved! I think such a result can be expected, may be it is more meaningful to concentrate on the conserved interactions than on the less conserved ones at higher concentrations, which might be not biological meaningful

Figure S8: a table specifying proteins that were detected or not in the presence of high MgCl₂ would be interesting

Line 262: shifts in melting temperature are another indirect indication of the structural changes upon metabolite addition. I was surprised that the authors did not validate the changed peptide coverage with the recombinant proteins in the LiP-SMap assay as done before with crude extracts?

Line 271: please add - rather affecting V_{max} but not K_m!

Line 286: please perform a statistical test with the activity data shown in Fig. S17. From the dots

probably representing single activity measurements, it seems likely that GAP is rather not stimulating the enzyme under reducing conditions!

Line 316: where are the data supporting this statement that 8 out of 13 metabolites affected SynTKT catalytic activity, the data shown in Fig. S18 for SynTKT show only an effect of AMP, all other curves +/- metabolite seem to be very close to each other

Line 381: this is true, but the LiP-SMap method basically depends on different conformations affecting proteinase action. Therefore, one should expect a better correlation here.

Line 413: glyoxylate is a rather strong acid and very reactive. Hence, it might have some direct effects on protein structures, unfortunately, there are no data on in vivo glyoxylate levels in cyanobacteria, but it should be quite low to prevent inhibitory effects. It is surprising that the authors did not find any effects by 2-phosphoglycolate (a proven indicator for low CO₂ availability in cyanobacteria), because it has been shown that 2PG negatively impacts Calvin cycle enzymes such as TPI and SBPase. Please discuss points like that.

Discussion: please discuss the rather low overlap between *Synechocystis* and *Synechococcus* regarding metabolite/protein interactions, because it is surprising in the light of the close similarity of the Calvin cycle enzymes

Reviewers' comments:

Reviewer #1 (Remarks to the Author):

This paper uses the technique limited proteolysis small molecule mapping (LiP-SMap) to explore the role of metabolite protein interactions in the Calvin cycle and to test the impact of these on the regulation of flux. The main findings of this work is that the regulation exerted by metabolites on the function of the Calvin cycle are more numerous than previously thought. The enzyme F/SBPase was shown to be differentially regulated by GAP dependent on the redox conditions.

The work presented in the paper is novel and will wide interest to the community outside of bacterial photosynthesis. The work presented is rigorous and the conclusions drawn are well balanced.

I think this paper will influence thinking in the field and forms a proof of concept piece of work for others to follow.

Overall this is a well crafted manuscript, that is written with clarity and presents areas for further work.

We thank the reviewer for their time and feedback.

Reviewer #2 (Remarks to the Author):

Sporre et al. performed a novel analysis based on proteolysis-coupled mass spectrometry to examine metabolite-protein interaction in cyanobacteria. Using four autotrophs, they performed LiP-SMap analysis and detected 8,000-15,000 peptides including those derived from Calvin cycle enzymes. They compared the peptide profiles in the presence of various chemicals and metabolites. The results showed the interaction between the Calvin cycle enzyme and metabolites such as glyoxylate, sugar phosphates, 2-OG, ATP, and so on. Finally, the authors validated the results by biochemistry determining kinetic parameters. Overall, this study is intriguing, and the novel methods contribute to the development of the understanding of cyanobacterial metabolisms. I have several comments before publication.

We thank the reviewer for their time and feedback.

Major points,

To validate the results of LiP-SMap analysis, the published biochemical data can be useful, but less cited in this paper. For example, citrate is a key metabolite regulating the enzymes of the oxidative pentose phosphate pathway, but not the TCA cycle (Ito and Osanai, 2020 Biochem J.). Also, various enzymes of primary carbon metabolism (Calvin cycle, OPP pathway, glycolysis, and TCA cycle) are known to be regulated by metabolites. Please cite the biochemical papers to validate the current results.

We thank the reviewer for the suggestion and agree with the reviewer that the overlap between LiP-SMap and known enzyme regulation should be more clear in the manuscript. We now reference some of the published data in the revised manuscript:

“L149: Recent in vitro characterization of enzymes from the oxidative pentose phosphate pathway of *Synechocystis* showed that several were unexpectedly inhibited by TCA-cycle metabolites 24,25.”

L310: Recent studies showed that citrate, among other TCA cycle intermediates, inhibited G6PDH and GND in *Synechocystis*, enzymes of the OPP pathway and isocitrate inhibited PRK^{24,25,45}. Comparison of LiP-SMap data with results from these studies showed moderate overlap with the reported enzyme regulations. Of four inhibitors of G6PDH, two were LiP-SMap hits (citrate, NADPH), and of two GND inhibitors, one was a LiP-SMap hit (citrate).

L673: In general, many more interactions were detected by LiP-SMap than via melting temperature shifts on purified F/SBPase and TKT enzymes. Comparison of LiP-SMap interactions to enzyme activity data, from published reports and those collected here, showed that fewer than half of LiP-SMap hits affect catalytic activity. Thus, LiP-SMap technique likely detects many interactions that alter proteinase K access but do not significantly alter

protein conformation; and most of the detected interactions have a effect on protein activity that could be described as fine-tuning.

Line 150, how do the authors determine the culture conditions for LiP-SMap? The culture conditions are important to compare the data and protein profiles are varied by cultivation conditions and growth phases.

We have updated the manuscript:

L181. "Proteins were extracted from cells growing in mid-log or late-log phase (OD730 approximately 1). For cyanobacteria, growth was autotrophic with 1% CO₂ at approximately 100 uE light. We have clarified in the text."

Minor points,
Line 28, sp. (non-italic and period)

This is fixed

Line 132, I cannot understand why the authors used *Cupriavidus necator* and *Hydrogenophaga pseudoflava* to compare with cyanobacteria. An additional explanation is required in the Introduction.

We have added some text motivating our use of *Cupriavidus* and *Hydrogenophaga* to the introduction

L169: "The strains are also of interest for biotechnology, as potential starting points for developing microbes to convert carbon dioxide into chemicals. Comparison of metabolite-protein interactions across the bacteria could reveal common metabolite-level regulation in Calvin-cycle based autotrophic metabolism. "

Line 175, readers are difficult to know why the authors tested the glyoxylate-treated cells for LiP-SMap analysis (I know photorespiration is mentioned in the Results, but until that, the readers cannot find it).

In this section, we chose glyoxylate treatment to test the reproducibility of the LiP-Smap results (two independent glyoxylate treatments done on different days). The reasoning for choosing glyoxylate was that it was an intermediate metabolite in terms of the number of interactions. We have clarified this in the text.

Line 220, please check the sl number for the gene encoding the cAMP receptor protein. Generally, sl numbers are four digits.

We fixed the locus number for SyCRP1 *sl1371*

Reviewer #3 (Remarks to the Author):

Review on MS COMMSBIO-23-0305: "Metabolite interactions in the bacterial Calvin cycle and implications for flux regulation", by Professor Hudson and colleagues

Modern high throughput methods made a great contribution to our understanding of biological phenomena on the systems level. In addition to changed transcript or protein abundances, the biochemical regulation of enzyme activities is a layer significantly contributing to flux alterations under different environmental conditions. Recently, the LiP-SMap method was published that permits a high throughput analysis of protein/metabolite interactions. The method has not been applied to the Calvin cycle, which was done here for the first time. CO₂ fixation through Calvin cycle is the main driver of the global carbon cycle. Moreover, microbial CO₂ fixation is the base for the future sustainable biotechnology, because it can use the greenhouse gas for multiple applications. The authors analyzed the potential regulation of Calvin cycle enzymes and others associated with this cycle in four different bacteria. Two cyanobacterial model strains were used, which are potentially interesting as so-called green cell factories and can also serve as model for the plant Calvin cycle. In addition, two heterotrophic bacteria were analyzed, which have the potential in biotechnology to produce organic compounds from CO₂ and H₂. In the beginning, the authors provided many data, which were obtained to evaluate the methods for its application to the specific question. As usual, the study discovered a great number of potential metabolite/enzyme interactions. The authors found that the Calvin cycle regulation due to different metabolites seems to be different between autotrophic and heterotrophic bacteria. In addition to the high throughput screening, some of the observed interactions were verified by subsequent detailed enzyme assays to characterize the biochemical and structural features. Finally, the impact of the newly discovered metabolic regulation of FBP/SBPase in *Synechocystis* was embedded in a model showing its impact on the overall Calvin cycle regulation.

We thank the reviewer for the summary and for their comments and suggestions.

Points to consider:

Line 167: please discuss if the partial oxidized state of the *Synechocystis* is rather because of oxidation through the protein extraction in the presence of oxygen or might reflecting the in vivo state of proteins!

The reviewer points out that proteins in the proteome extract could be partially oxidized because of the cellular state, or because of partial oxidation that occurs during extraction, digestion, and LC-MS. We were not aware that spontaneous oxidation was so pervasive in proteomics workflows. In the revised manuscript Results section, we mentioned that because of this oxidation we cannot determine

the cellular redox state from proteomics. Many oxidation paths that occur in MS are likely in common for non-DTT and DTT treated extracts, since these oxidation events occur after trypsin digestion and column purification of peptides. Some oxidation paths, such as during trypsin digestion, could be affected differently by the presence of DTT. We can attempt to distinguish between these two possible effects by comparing the proteins affected by DTT to known redox regulated proteins in *Synechocystis*.

L201: A recent proteomics study identified 611 redox-sensitive proteins in *Synechocystis*³⁶; 84% of the DTNB-affected proteins and 25% of the DTT treated proteins were among those reported as redox regulated. Since the *Synechocystis* proteome extracts were sensitive to both DTT and DTNB, the proteins may be partially oxidized in the cell at the harvesting condition. However, peptide oxidation can occur at many steps throughout the proteomics workflow³⁷, which complicates estimates of the extent of reduction in a given cultivation conditions.

Line 171: please mention the overall coverage of the proteome of the 4 bacteria

This is now reported in the Results section

L226: Maximum protein counts were 1896 for *Synechocystis*, 1682 for *Synechococcus*, 2032 for *Cupriavidus*, and 1752 for *Hydrogenophaga*.

Line 178: I guess that this number relates to overall peptide detection (which is surprisingly low), it might be better to compare the values here to the reproducibility of detected metabolite/peptide interaction LiP_SMAP experiments in *E. coli* or yeast

We do think that the modest overlap (35%) of significantly affected peptides in the two separate glyoxylate treatments is due to a low proteome coverage. For example, we detect 1682 proteins in *Synechococcus* extracts, while the original LiP-SMAP paper of Piazza et al *Cell* 2018, detected 2500+ proteins in *E. coli* extracts. It is likely that many peptides in our samples have a weak signal, and thus more variance, so that they do not meet the significance threshold. Interestingly, we did not find any reporting of similar “independent tests,” of the reproducibility of significant peptides when treated with a test metabolite in previous Lip-SMAP papers.

Line 194: The data in Fig. S5 seem to differ from the data shown in Fig. 1B. A visual comparison of red or blue peptide marks in high versus low concentrations indicates a much lower interaction with GAP but also AcCoA between high and low concentration, please clarify this

In Figure S5, the number of proteins with significant interactions are shown. In Figure 1B, we are showing peptides that are affected (often multiple per protein). We checked the figures and they appear to be consistent.

Line 212: does this mean that the more specific interactions, which can be assumed to remain visible when adding lower metabolite concentrations, are more conserved! I think such a result can be expected, may be it is more meaningful to concentrate on the conserved interactions than on the less conserved ones at higher concentrations, which might be not biological meaningful

It makes sense that low-concentration interactions are those of high affinity. Whether these are more evolutionary conserved is difficult to say. Part of the challenge of this study was selecting the appropriate concentrations. In some cases (e.g. L-Phe treatment), known biological interactions only show up in the high-treatment concentrations. The sparsity of detected interactions in the Low concentration treatment could be due to testing too low concentration.

Figure S8: a table specifying proteins that were detected or not in the presence of high MgCl₂ would be interesting

These are now provided as Supplemental Dataset S7

Line 262: shifts in melting temperature are another indirect indication of the structural changes upon metabolite addition. I was surprised that the authors did not validate the changed peptide coverage with the recombinant proteins in the LiP-SMap assay as done before with crude extracts?

This is an interesting follow-up that we have now included in the revised manuscript. We performed Lip-SMAP on the purified syn-F/SBPase with added GAP and NADPH.

L458: We also screened for interactions from added GAP and NADPH on the purified, recombinant syn-FBPase. We detected approximately 40 peptides, and protein coverage was >90%. Addition of GAP and NADPH resulted in 4 and 6 significantly altered peptides ($q < 0.05$), respectively, more than were found in the proteome extract LiP-Smap (2 for GAP and 3 for NADPH). Nevertheless, altered peptides from the purified enzyme included those altered in the proteome extract LiP-Smap for both metabolite treatments, indicating that the LiP-Smap hits for these metabolites in proteome extracts were not due to conversion of the metabolite (Figure S15).

Figure S15. Comparison of affected peptides from LiP of syn-F/SBPase for purified protein and from proteome extracts. A) Highlight of peptides (colored green) significantly affected in a LiP experiment with GAP treatment (0.5 mM), purified syn-F/SBPase. B) Highlight of peptides (purple) affected in Lip experiment with GAP treatment (0.5 mM), syn-F/SBPase in *Synechocystis* proteome extracts. C) Highlight of peptides (green) affected in Lip experiment with NADPH treatment (3 mM), purified syn-F/SBPase. D) Highlight of peptides (purple) affected in Lip experiment with NADPH treatment (3 mM), syn-F/SBPase in *Synechocystis* proteome extracts. Syn-F/SBPase structure is the monomer from PDB 3RPL. The FBP substrate is in sticks and Mg^{2+} are in yellow spheres.

Line 271: please add - rather affecting V_{max} but not K_m !

We checked again the fits for the kinetic data of F/SBPase with added GAP. Here, the significant effect is on K_m , while differences in V_{max} are not significant. That is, GAP stimulates activity of F/SBPase only at low FBP concentrations.

Line 286: please perform a statistical test with the activity data shown in Fig. S17. From the dots probably representing single activity measurements, it seems likely that GAP is rather not stimulating the enzyme under reducing conditions!

We now include a statistical test for GAP stimulation via LC/MS. As the reviewer suggests, the stimulation is not statistically significant ($p = 0.44$), due to variability among replicates. Unfortunately, we were not able to run additional assays due to

technical difficulties with the machine. However, we now clarify that the LC/MS experiment was included to verify that trends observed with Malachite Green assay (Pi detection) were not due to GAP interacting with MG reagents. Since the results of the MG assay and LC/MS were similar in trend, we think that the MG assays present in Figure 4 are an accurate measure of the F/SBPase reaction rate. Therefore, the totality of our data suggest that GAP stimulates F/SBPase activity, even if the difference was not significant in the specific experiment where we also used LC/MS. To clarify, we now include the following statement in the Results section:

L544. To exclude the possibility that GAP affected the enzyme assays by interacting with MG or other components, we also measured the F6P product from some syn-F/SBPase reactions via LC-MS. The observed trends (GAP stimulation with DTT, and inhibition in absence of DTT) were similar to those observed in the MG assays, though variation among replicates resulted in non-significant differences for GAP activation (**Figure S17**).

Line 316: where are the data supporting this statement that 8 out of 13 metabolites affected SynTKT catalytic activity, the data shown in Fig. S18 for SynTKT show only an effect of AMP, all other curves +/- metabolite seem to be very close to each other

We have added a new Supplemental Table S4 showing the kinetic parameters of TKT for these treatments, which are calculated from the fits of Figure S8. Significant ($p < 0.05$) difference to untreated enzyme is indicated. From this new analysis we now report that 5/13 metabolites affected activity of syn-TKT, and 3/10 affected activity of cn-TKT. We have also edited the Results section with these new values.

“L556. Fewer than half of the TKT-interacting metabolites detected by LiP-SMap altered TKT catalytic activity *in vitro* (5/13 for *syn*-TKT and 3/10 for *cn*-TKT), and only a few affected kinetic parameters by more than 20% or significantly affected T_m in thermal shift assays (**Table S4**).”

Significant effects on V_{max} of syn-TKT: AcCoA, ATP, G6P, KDGP
Significant effects on K_m of syn-TKT: AMP

Significant effects on V_{max} of cn-TKT: none
Significant effects on K_m of cn-TKT: 2PG, DHAP, NADP

Line 381: this is true, but the LiP-SMap method basically depends on different conformations affecting proteinase action. Therefore, one should expect a better correlation here.

We now add to this section that LiP-SMAP can also detect interactions that simply block PK access to a protein, as well as conformational changes. In the case of

conformational changes, it may also be possible that LiP is a more sensitive detection than melting-temperature analysis.

L676 “In general, many more interactions were detected by LiP-SMap than via melting temperature shifts on purified F/SBPase and TKT enzymes. Comparison of LiP-SMap interactions to enzyme activity data, from published reports and those collected here, showed that fewer than half of LiP-SMap hits affect catalytic activity. Thus, LiP-SMap technique likely detects many interactions that alter proteinase K access but do not significantly alter protein conformation; and most of the detected interactions have an effect on protein activity that could be described as fine-tuning.”

Line 413: glyoxylate is a rather strong acid and very reactive. Hence, it might have some direct effects on protein structures, unfortunately, there are no data on in vivo glyoxylate levels in cyanobacteria, but it should be quite low to prevent inhibitory effects. It is surprising that the authors did not find any effects by 2-phosphoglycolate (a proven indicator for low CO₂ availability in cyanobacteria), because it has been shown that 2PG negatively impacts Calvin cycle enzymes such as TPI and SBPase. Please discuss points like that.

In the revised manuscript we apply a significance cutoff of $q < 0.05$ for plotting interactions. This results in revised Figure 2 and Figure 3 (and Figures S7 and S9), where $q < 0.01$ was considered previously.

Revised Figure 3, showing interactions with $p < 0.05$

Revised Figure 2, where interactions with $p < 0.05$ are included in analysis

Examining 2-PG interactions with *Synechocystis*, we find 40+ peptide interactions with 2-PG in *Synechocystis* ($q < 0.05$). We now add a section on this to the Discussion

L712. “Phosphoglycolate (2-PG), also a photorespiratory metabolite, may serve as a general carbon status indicator in *Synechocystis* and trigger transcriptional expression of carbon uptake (Haimovich-Dayan et al. 2015). In *Synechocystis*, 2-PG (high concentration addition) interacted with multiple ribosomal subunits, and ribosome hibernation and elongation factors, suggesting a direct effect on protein translation. Other interactions were among enzymes in nucleic acid synthesis (PyrE, PyrH, Slr1616 and Slr1619), a phosphate transporter (PtsS), and ATPase subunits AtpA and AtpB. In *Arabidopsis*, 2-PG negatively affected activities of Calvin cycle enzymes SBPase and TPI (Flügel et al. 2017). SBPase is not present in bacteria, and TPI was not among affected proteins in our study. However, TAL and FBA did have interactions ($p < 0.05$) to 2-PG in *Synechocystis*.”

Discussion: please discuss the rather low overlap between *Synechocystis* and *Synechococcus* regarding metabolite/protein interactions, because it is surprising in the light of the close similarity of the Calvin cycle enzymes

We have modified the Discussion section to include this point. We think that the relatively low overlap between *Synechocystis* and *Synechococcus* is mostly due to

the undersampling of the MS-based proteomics, combined with the variability in the Lip-SMAP with regards to calling significant hits. This means that we may not detect the same peptides in enzyme orthologs, and that some interactions do not meet significant cutoff despite substantial changes in abundance. In the revised manuscript, we consider a more relaxed significance cutoff ($p_{\text{adj}} < 0.05$), and here the correlation between *Syn6803* and *Syn7942* is improved (improved clustering between *Syn6803* and *Syn7942*). As the reviewer points out, these strains have high homology among Calvin cycle enzymes (e.g. 76% identify for Tkt and 80% identify for F/Sbpase, for example).

L777: “As a result, even some known metabolite regulators will not be detected by LiP-SMap. The high variability in peptide coverage and in peptide quantification likely contributes to the low overlap in significant interactions among central carbon enzymes of *Synechocystis* and *Synechococcus*, even though these enzymes have high sequence homology. While PCA analysis of all interactions showed the cyanobacteria clustered together for some metabolites, the species discrimination axis had a weight of only 6%.”

REVIEWERS' COMMENTS:

Reviewer #2 (Remarks to the Author):

The authors answered all the comments from the reviewers. I agree with the publication.

Reviewer #3 (Remarks to the Author):

I thank the authors for their efforts during revision. I have no new critical comments.